# PURE-seq integrates FACS and PIP-seq for single-cell genomics of ultra-rare cells

Sixuan Pan [1,11], Inés Fernández-Maestre[2,3,11], Kai-Chun Chang [1],
Stéphane Van Haver[4,5], Matthew G. Wereski[2], Alexandra M. Haugh[6], Katy K. Tsai[6],
Adil I. Daud [6], Robert L. Bowman [7], Harish N. Vasudevan [8],
Ross L. Levine [2,9,10] ✉ & Adam R. Abate [1] ✉

Single-cell transcriptomics is valuable for uncovering individual cell properties, particularly in heterogeneous systems. However, this technique often results in the reanalysis of many well-characterized cells, increasing costs and diluting rare cell populations. To address this, we develop PIP-seq for Rare-cell Enrichment and Sequencing (PURE-seq). PURE-seq allows direct FACS sorting of cells into PIP-seq reactions, minimizing handling and reducing cell loss. PURE-seq reliably sequences ultrarare cells, with 1 hour of sorting capturing tens of target cells at a rarity of 1 in 1,000,000. Leveraging this extreme sensitivity, we use PURE-seq to isolate and single-cell sequence circulating tumor cells from metastatic melanoma patient blood, obtaining detailed single cancer cell gene expression profiles. Additionally, we use PURE-seq to examine hematopoietic stem and progenitor cells from young, old and middle-aged mice. Transcriptomic analysis identifies *Egr1* as a putative master regulator of murine hematopoietic stem and progenitor cell aging, demonstrating PURE-seq's utility as a discovery platform for basic science applications. PURE-seq offers a simple and highly sensitive method for single-cell sequencing ultra-rare cells.

Rare cells often have critical biological roles and substantial clinical relevance, yet their low abundance makes them challenging to study. Circulating tumor cells (CTCs) exemplify this problem: originating from primary or metastatic tumors, CTCs enter the bloodstream, seed metastases, reflect tumor evolution in real-time, and correlate with treatment response and patient survival[1,2]. Therefore, analyzing CTCs can greatly enhance our understanding of metastasis, inform therapeutic strategies, and enable non-invasive disease monitoring.

However, because CTCs are exceptionally rare—often fewer than one per million blood cells—capturing and sequencing them efficiently remains difficult[1,2]. Conventional single-cell sequencing approaches are inefficient in such scenarios, dedicating most sequencing resources to abundant cell populations while missing rare but biologically significant cells[3]. Ideally, one would enrich rare cells with high fidelity before sequencing, thus ensuring the efficient and meaningful use of sequencing capacity.

[1]Department of Bioengineering and Therapeutic Sciences, University of California San Francisco, San Francisco, CA, USA. [2]Human Oncology and Pathogenesis Program, Memorial Sloan Kettering Cancer Center, New York, NY, USA. [3]Louis V. Gerstner Jr Graduate School of Biomedical Sciences, Memorial Sloan Kettering Cancer Center, New York, NY, USA. [4]Molecular Pharmacology Program, Memorial Sloan Kettering Cancer Center, New York, NY, USA. [5]Tow Center for Developmental Oncology, Memorial Sloan Kettering Cancer Center, New York, NY, USA. [6]Department of Medicine, Division of Hematology/Oncology, University of California San Francisco, San Francisco, CA, USA. [7]Department of Cancer Biology, Perelman School of Medicine, University of Pennsylvania, Philadelphia, PA, USA. [8]Departments of Radiation Oncology and Neurological Surgery, University of California San Francisco, San Francisco, CA, USA. [9]Department of Medicine, Weill Cornell Medical College, New York, NY, USA. [10]Center for Hematologic Malignancies, Memorial Sloan Kettering Cancer Center, New York, NY, USA. [11]These authors contributed equally: Sixuan Pan, Inés Fernández-Maestre. ✉e-mail: leviner@mskcc.org; arabate@gmail.com

Common methods for enriching rare cells include antibody-based approaches like fluorescence-activated cell sorting (FACS), magnetic-activated cell sorting (MACS), and cell levitation, which isolate cells based on specific surface markers[4–6]. However, these enrichment methods do not integrate seamlessly with existing single-cell sequencing platforms, requiring manual transfer steps that risk cell loss or degradation, a significant concern when handling ultra-rare populations. An optimal workflow would directly combine enrichment and sequencing steps in a single, streamlined process, eliminating manual handling and preserving rare cells.

Here we introduce PURE-seq (PIP-seq for Rare-cell Enrichment and Sequencing), a method that integrates FACS with Pre-templated Instant Partition sequencing (PIP-seq)[7]. PIP-seq enables single-cell RNA sequencing within a standard 0.5 mL Eppendorf tube, the exact tube type commonly used by flow cytometers, allowing direct sorting of cells into the sequencing reaction. A brief vortex encapsulates cells into droplets, immediately lysing them and initiating barcoding without any additional handling. This simplified approach enables a one-hour sorting run to capture tens of cells at frequencies as low as 1 in a million.

We validated PURE-seq using two challenging experimental systems. First, we enriched CTCs from metastatic melanoma patients through CD45-depletion, enabling unbiased transcriptional profiling of CTC heterogeneity. Second, we sequenced murine hematopoietic stem and progenitor cells (HSPCs), which are challenging to capture in sufficient numbers with conventional methods, to explore changes related to aging[8,9]. Our results identified elevated expression of the gene *Egr1* in older cells, implicating it as a potential master regulator of murine HSC aging, consistent with observations in human hematopoietic stem and progenitor cells[10–12]. In both scenarios, PURE-seq generated high-quality single-cell transcriptome data from very limited cell inputs, reproducing results comparable to gold-standard methods, while offering a simpler, more efficient, and far more scalable workflow.

## Results

### PURE-seq workflow

The PURE-seq workflow utilizes readily available commercial platforms, FACS and PIP-seq, to achieve scalable, reliable, and accessible sequencing of ultra-rare cells. In PURE-seq, cells are sorted directly into single-cell barcoding reaction tubes. Subsequent cell encapsulation follows the standard PIP-seq protocol[7], which involves adding encapsulation oil, vortexing for one minute, lysing cells, and capturing mRNA (Fig. 1A). To optimize cell viability and capture efficiency, we fine-tuned cell sort stream alignment, sorting speed, and total sorting duration ("Methods").

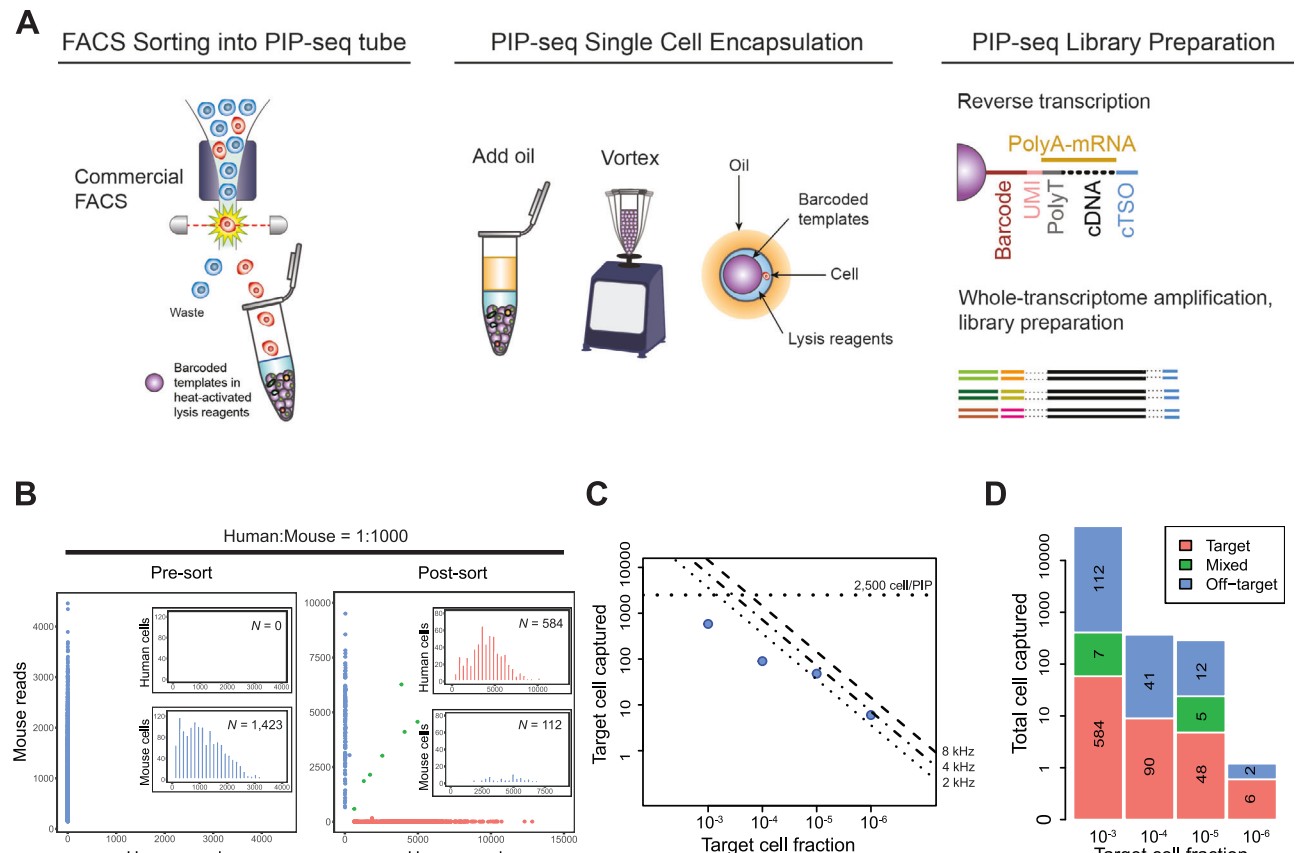

**Fig. 1 | PURE-seq efficiently captures and sequences rare cells isolated by FACS. A** PURE-seq utilizes a commercial FACS system to sort target cells directly into PIP-seq reaction tubes containing barcoded templates in heat-activated lysis reagents. The subsequent single-cell encapsulation in droplets follows the standard PIP-seq protocol[7], which involves adding oil, vortexing, heat-activated lysis, and capturing mRNA on the barcoded templates. After mRNA capture, reverse transcription, and whole-transcriptome amplification are conducted in bulk to prepare barcoded cDNA for Illumina sequencing. **B** Barnyard plots of mixed human-mouse population (Human:Mouse = 1:1000) sequenced before (left) and after sorting (right). Inserts are histograms of read distribution for sequenced human or mouse cells. Cells are colored by cell type (blue, mouse reads; red, human reads; green, mixed reads). **C** Number of target cells captured as a function of target cell fraction. The dashed lines mark the theoretical limit of the captured cells. A maximum of 2500 cells is sorted into each T2 PIP-seq reaction. Contour lines are the theoretical numbers of target cells that can be sorted within 60 min with different sorting rates (8 kHz, 4 kHz, and 2 kHz). Blue dots are the actual number of cells sequenced for the mixed human-mouse population with target cell fractions of $10^{-3}$, $10^{-4}$, $10^{-5}$, and $10^{-6}$. **D** Number of target and off-target (mis-sort) cells sequenced for each rarity group.

Fluorescence-activated cell sorters offer multiple sorting precision modes. In the single-cell mode, the sorter prioritizes specificity, discarding ambiguous events arising from staining variability, cell clumping, or detector coincidences. In contrast, the yield mode captures these ambiguous events to maximize recovery of rare cells, even if this leads to capturing some off-target cells. PURE-seq leverages the substantial downstream sequencing capacity of PIP-seq—capable of processing inputs of 2000, 20,000, or even over 100,000 cells—to prioritize capturing rare cells at the expense of purity. Since all sorted cells undergo single-cell sequencing, off-target cells can be readily identified and removed through computational clustering. This approach ensures maximal recovery of ultrarare target cells with only a modest increase in sequencing reads dedicated to off-target populations. Consequently, PURE-seq achieves substantial enrichment from pre-sort to post-sort while maintaining practical sequencing efficiency[7].

To assess the efficacy of PURE-seq, we performed a species-mixing experiment by spiking human HEK 293 T cells into mouse NIH 3T3 cells at a ratio of 1:1,000. The human (HEK 293 T) cells represented the rare target population within an abundant mouse background. We labeled the human cells with Calcein Red-Orange dye and the mouse cells with Calcein Green dye ("Methods"), then processed the mixed sample using the BD FACS Aria system. As an unenriched control, we sorted the first 2500 cells (regardless of species) into the PIP-seq reaction. In parallel, as the enriched sample, we specifically sorted the first 2500 human (red-orange) cells into a separate PIP-seq reaction. After barcoding and sequencing, both samples produced high-quality single-cell data, as demonstrated by clear separation along the human and mouse axes of the barnyard plots (Fig. 1B). While the unenriched control captured no human HEK 293 T cells, the enriched sample (sorted in yield mode) successfully recovered 584 human HEK 293 T cells along with 112 off-target mouse cells. These results confirm that PURE-seq reliably enriches and sequences rare target populations at frequencies as low as 1:1000.

A major strength of flow cytometry is its capacity for high-throughput sorting, enabling the screening of large populations to detect and isolate rare cells. We thus sought to determine the maximum rarity of target cells that could be reliably captured and sequenced using PURE-seq. To achieve this, we optimized sorting parameters to maximize total sorted cells while attempting to minimize potential perturbation of gene expression caused by extended sorting times and shear stress. Therefore, we limited sorting duration to a maximum of 60 min and sorting speed to 8 kHz, allowing ~28.8 million cells to be processed per run. With these conditions, FACS can deliver tens of target cells at a frequency of ~1 in 1 million to the PIP-seq reaction.

Achieving successful sequencing from such small numbers of cells requires exceptionally efficient barcoding, as commercial single-cell platforms typically require hundreds or thousands of input cells. Given that the maximum input volume for the PIP-seq T2 kit is 5 μL, we limited the maximum number of sorted cells to 2500, based on the BD FACS Aria's droplet volume of 1.81 nL per drop. If more sorted cells are desired, larger-capacity PIP-seq kits (e.g., T20 for 20,000 cells, or T100 for 100,000 cells) can be employed. We systematically evaluated the enrichment capability of PURE-seq by sorting target cells at varying frequencies ranging from $10^{-3}$ to $10^{-6}$ (Fig. 1C, Supplementary Fig. 1A–D). Across this frequency range, we successfully captured and sequenced between 564 and 6 target cells, respectively, with purities exceeding 75% (Fig. 1D). Switching sorting to purity mode increased purity to 98% but reduced cell recovery by 33% (Supplementary Fig. 1E, F).

## PURE-seq profiles melanoma circulating tumor cells

Following validation of PURE-seq's technical performance, we applied the method to analyze circulating tumor cells (CTCs) from peripheral blood samples of patients with metastatic melanoma (n = 2). CTCs shed from primary and metastatic tumors circulate in the bloodstream and provide a minimally invasive resource that can inform diagnosis, monitor disease burden, and guide treatment decisions[13,14]. However, their extreme rarity and phenotypic heterogeneity impede comprehensive transcriptomic analysis. PURE-seq could circumvent these barriers by enabling sensitive, unbiased single-cell expression profiling. We isolated peripheral blood mononuclear cells (PBMCs) from 8 mL whole blood per patient and sorted CD45- cells directly into PURE-seq reactions (Fig. 2A). Conventional CTC isolation methods in melanoma rely on tumor-specific cell surface markers such as EpCAM or MCSP and therefore miss cancer cells lacking such markers[15,16]. To avoid this bias associated with positive selection approaches, we employed a negative enrichment strategy that depletes CD45+ leukocytes, thereby enriching non-hematopoietic cells without presuming tumor-associated antigen expression.

For identification of CTCs in the sequenced data, we pooled CD45+ (control) and CD45- (negative) samples from the two patients for an integrated analysis (Fig. 2B). Unsupervised clustering of 896 cells revealed three distinct clusters: a T/NK cell cluster highlighted by expression of the hematopoietic lineage marker *PTPRC* (the gene encoding CD45) and other T/NK markers (*IL7R, S100A4, GNLY, NKG7*), and two CTC-enriched clusters lacking *PTPRC* expression highlighted by expression of reported melanoma-associated marker genes from the Jensen Diseases database[17] (*KIAA1217, SPAG17, CSMD2, KALRN, SDK1, IL36B, PLCL2, SLIT3, GRIN2B, COL19A1*) (Fig. 2C, D, Supplementary Fig. 2A, and Supplementary Data 1). To rule out other non-tumor, CD45- cell populations found in peripheral blood such as platelets, endothelial and fibroblasts, we evaluated expression of marker genes for each (platelets: *PPBP, TUBB1, PF4*; endothelial: *VWF, PECAM1, FLT1, ICAM1*; fibroblasts: *FAP, PDPN, COL1A1*) and found none were expressed in either cluster 1 or 2 (Fig. 2D).

Interestingly, the two CTC-enriched clusters exhibited overlapping and distinct melanoma-associated gene expression, revealing heterogeneity within the rare CTC population. Both clusters were specific to the melanoma disease type, underscoring their malignant status (Supplementary Fig. 2B, Supplementary Data 2, 3). Gene Set Enrichment Analysis (GSEA) further revealed distinct upregulation of the KRAS signaling DN signature in cluster 2 CTCs, suggesting potential differences in KRAS pathway modulation and further underlying CTC heterogeneity (Supplementary Fig. 2C, Supplementary Data 4–8). Additionally, comparison of differentially expressed genes (DEGs) between CTCs and hematopoietic cells (CD45+) indicated higher levels of genes expressed in CTCs, such as *NPAS2*[18], *IDO1*[19], *SVIL*[20], and *SPAG17*[21], all of which have been previously associated with melanoma (Supplementary Fig. 2D, E). Importantly, these CTCs lacked expression of canonical epithelial or melanoma lineage markers commonly targeted for positive enrichment (*EPCAM, MITF, MCAM, CSPG4, PRAME*), indicating that these cells would likely have been missed by conventional marker-based methods[15].

To independently verify the malignant status of these cells, we used the CD45+ sorted cells in clusters 1 and 2 as reference cells for inferCNV analysis, which revealed numerous copy number variations in the CD45- sorted cluster 1 and 2 cells, supporting their designation as CTCs (Fig. 2E). We observed CNV patterns consistent with metastatic melanoma. Sample 1 exhibited amplifications on chromosomes 6q, 13, and 20, consistent with frequent 20q gains reported to occur early in melanoma tumorigenesis and potentially initiating cancer progression[22]. Sample 2 presented a broader spectrum of chromosomal gains, including 2p, 3, 5, 10p, and 18, accompanied by losses on 6p, 11, 15, 19, and 22. Notably, loss of chromosome 11 is a common event in melanoma and has been linked to advanced tumor stage and poor prognosis[23–25]. Overall, the distinct CNV landscapes reflect tumor heterogeneity and provide further validation for the malignant identity of the CTC populations we have isolated[26].

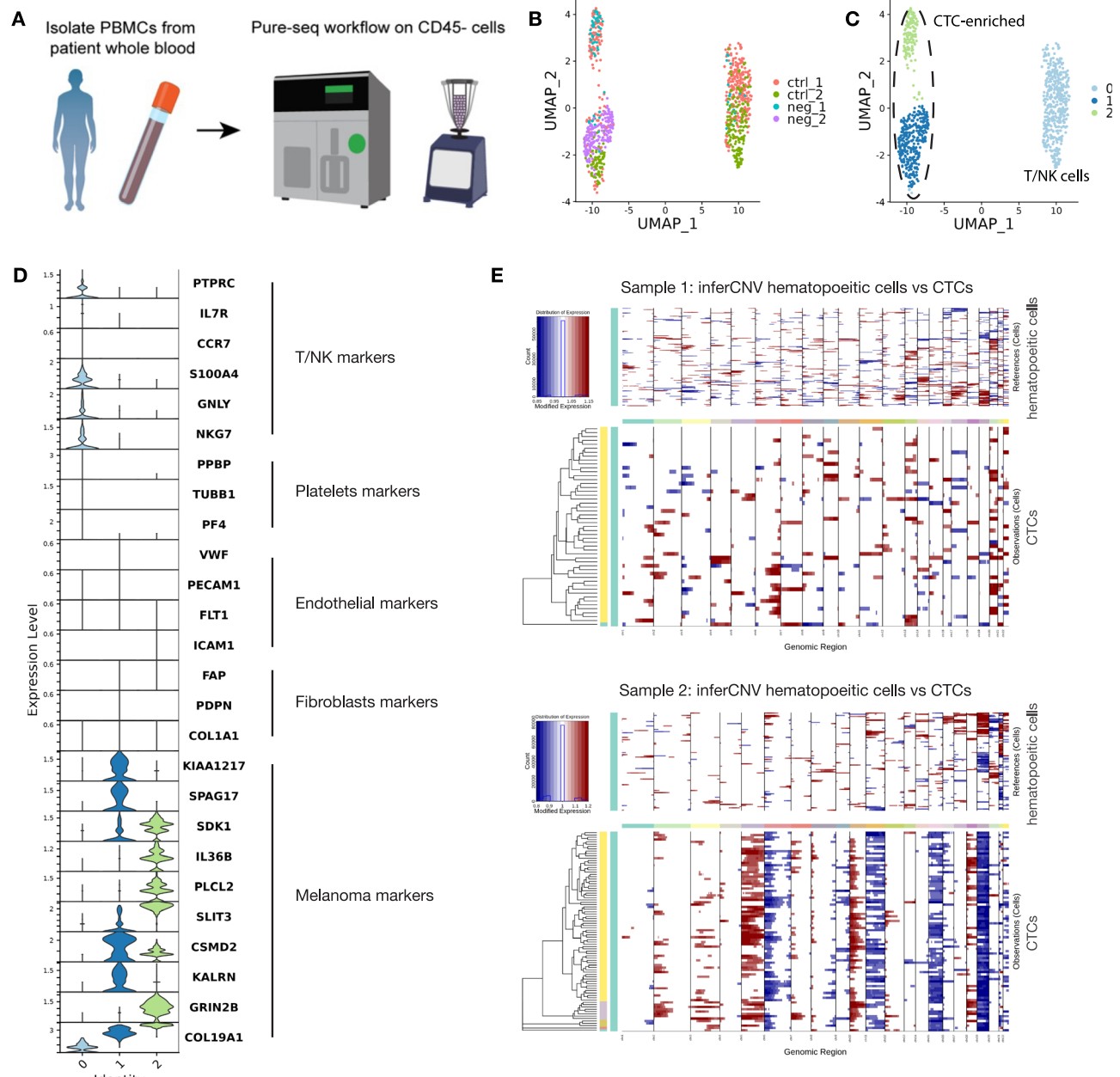

**Fig. 2 | PURE-seq profiles circulating tumor cell from melanoma patient blood samples. A** Blood samples from metastatic melanoma patients undergoes PBMC isolation followed by CD45- cell sorting into PURE-seq reactions (n = 2 patients, 8 mL blood each). **B** UMAP plot of all cells (n = 896) from ctrl (CD45+) and neg (CD45−) sorted samples. **C** UMAP identifies 3 clusters, which include 1 T/NK cell cluster and 2 CTC-enriched (dotted line). **D** Violin plots of expression levels of canonical genes (T/NK, platelets, endothelial, fibroblast, melanoma) in the different clusters. **E** InferCNV analysis of CTCs and reference hematopoietic cells by each patient.

As a further validation, integration of a publicly available healthy PBMC scRNA dataset (n = 2563 cells) with the PURE-seq dataset demonstrated that the identified CTCs clustered together, while the hematopoietic cells from the PURE-seq dataset clustered with the healthy PBMC dataset (Supplementary Fig. 3F, H). Collectively, these results demonstrate the ability of PURE-seq to isolate and profile rare CTCs from peripheral blood using a negative enrichment strategy, identifying two distinct melanoma CTC clusters that reflect the heterogeneity of metastatic melanoma at high resolution.

**PURE-seq significantly enriches murine HSPCs**
As another demonstration of PURE-seq for rare cell applications, we applied it to study hematopoietic stem and progenitor cells (HSPCs).

Profiling rare HSPC populations in scRNA-seq studies has been especially challenging due to their rarity and heterogeneity, which makes it difficult to capture enough true LT-HSCs for detailed analysis[8,9,27]. In a simplified sorting strategy (Supplementary Fig. 3A), we sorted from Lineage−Sca-1+c-Kit+ (LSK) cells based on the expression of SLAM markers to enrich for CD150+CD48− LSK cells. Specifically, to demonstrate how PURE-seq can increase the capture of HSPCs compared to a pre-sort control and provide a high-quality dataset to gain biological insights, we studied LT-HSCs throughout murine aging. We harvested whole BM cells from young (2-3 months old), middle-aged (12−14 months old), and old (18−20 months old) C57BL/6 mice. To enrich for HSPCs, lineage-positive cells were depleted using magnetic-activated cell sorting (MACS), after which Sca1+Kit+CD150+CD48− cells

were sorted from the pooled BM samples (n = 2–3 mice per pool) using the PURE-seq workflow. (Fig. 3A). To robustly identify the target cell population, we applied scGate[28], a marker-based purification algorithm previously benchmarked on scRNA-seq datasets that efficiently isolates defined cell populations of interest. Our analysis revealed that 51.8% of captured cells were Sca1+Kit+Flt3−Slamf1+CD48− (Fig. 3B) (of note, *Slamf1* encodes for the phenotypic cell surface marker CD150[29]).

We observed that these HSPCs exhibited near-absent expression of *Cd48* (Fig. 3C), consistent with our FACS gating strategy, which excluded CD48+ cells from all pooled samples (Supplementary Fig. 3A, B). *Flt3* expression was similarly nearly undetectable, in contrast to the clear expression of *Ly6a* (encoding Sca-1), *Kit* (c-Kit), and *Slamf1* (Fig. 3C). Our scGate analysis by age group revealed that the percentage of LT-HSCs increased with age (Fig. 3D), which aligns with previous studies demonstrating an increase in their percentage within the aged BM[30,31]. HSPCs cells noticeably co-localized with the expression of key LT-HSC genes such as myeloproliferative leukemia virus oncogene (*Mpl*), endoglin (*Eng*), MDS1 (*Mecom*), Meis homeobox 1 (*Meis1*), and homeobox genes (*Hoxb4* and *Hoxb5*) (Fig. 3E).

As a control, we sequenced pre-sort samples using the PIP-seq pipeline and found that only 0.54% of the cells were identified as HSPCs cells by scGate, indicating that with PURE-seq, we were able to enrich the percentage of LT-HSCs by 96-fold (51.8% vs 0.54%). Regarding the pre-sort control, we also detected that even though the samples were enriched for HSPCs, there were still differentiated immune cells and non-hematopoietic BM cell types, such as endothelial cells and fibroblasts (Supplementary Fig. 3C), which highlights the inefficiency of cell enrichment methods, such as MACS, for lineage-positive hematopoietic cell depletion (as we used in our experiment). In terms of the post-sort samples, 6725 cells that passed the Seurat quality control were captured, with an average of 841 cells per sample after sorting 2500 cells with the single-cell mode (Supplementary Fig. 3D). This demonstrates that 33.64% of the sorted cells were of high quality, a percentage that can be increased using the yield mode, as shown in our sorting precision modes experiment (Supplementary Fig. 1E, F). While we employed a basic gating strategy based on expression of SLAM family markers (CD150+, CD48−), which has also been recently used by others for sorting HSPC populations for single-cell genomics[32], additional markers can be incorporated to refine the isolation of functional HSPCs. For instance, CD49b expression distinguishes myeloid-biased (CD49b−) from lymphoid-biased (CD49b+) HSPCs and has recently been shown to remain conserved with aging[33]. Further, the inclusion of CD34 and CD135 (encoded by *Flt3*) as negative selection markers can further enrich for hematopoietic stem cells (HSCs), which are characterized by a CD34−CD135− phenotype[34].

After integrating all the samples, we unbiasedly identified 12 clusters based on transcriptomic differences (Supplementary Fig. 3D–H, Supplementary Data 9). Next, we conducted an integrated analysis with a publicly available dataset from Héuralt et al[35]. Similarly, they analyzed LT-HSCs from pooled FACS-sorted bone marrow samples of young and old mice by first depleting lineage-positive cells and then sorting Lineage (Lin)−Sca-1+c-Kit+Flt3− HSPCs, but using the 10x Genomics platform instead. They characterized their cell clusters based on differential gene expression analysis in combination with gene set enrichment analysis and gene signatures related to hematopoiesis. Based on their gene markers, we were able to identify 14 out of their 15 signatures (Supplementary Data 10, 11), mostly coinciding with non-primed signature clusters, thus classified because of their lack of expression of lineage-restricted genes (i.e., interferon response (ifn), non-primed (np)2, growth factor signaling (tgf), np4, replicative (rep), and dividing (div)). These non-primed signature clusters were in the head of the UMAP plot (Fig. 3F). The proportion of HSPCs was indeed higher in non-primed clusters compared to the primed ones (Fig. 3G). We identified five lineage-primed clusters enriched for cells expressing gene markers associated with lymphocyte (pL1), megakaryocyte

(pMk), neutrophil (pNeu), mastocyte (pMast), and erythroid (pEr) commitment. Notably, pL1, pNeu and pMast were located at the distal end of the trajectory tail, while pEr formed a completely distinct cluster, spatially separated from the main cell population (Fig. 3F). The proportion of HSPCs within these primed clusters was lower than 15% (Fig. 3G, Supplementary Data 12).

Our dataset was largely comparable to datasets generated with 10X Genomics Chromium, with a predominance of non-primed hematopoietic cell clusters[35], as expected due to the inclusion of SLAM markers in our FACS gating strategy (Supplementary Fig. 3A). Furthermore, we integrated our dataset with that of Héuralt et al[35]. (Supplementary Fig. 4A–C) and observed similar gene expression patterns across corresponding clusters, as shown in the DEG heatmaps (Supplementary Fig. 4D). We compared gene expression profiles by cluster of the non-primed cells and all clusters exhibited a correlation greater than $R^2 = 0.7$ (Supplementary Fig. 4E). We also compared the expression of key HSC-associated genes across both datasets using UMAP visualization (Supplementary Fig. 4F), which showed similar expression patters. Notably, however, our dataset contained a higher proportion of HSPCs compared to theirs (51.8% vs. 28.6%). In addition, the good quality metrics across our 12 unbiasedly identified clusters (Supplementary Fig. 3F), comparable to the Héuralt et al dataset upon integration (Supplementary Fig. 4B), the clear split by biological condition (i.e., age group) with concomitant detection of differences in cell numbers across clusters in our integrated dataset (Supplementary Fig. 3G), indicated the suitability of PURE-seq as a reliable alternative pipeline to isolate a rare cell population and analyze their single-cell transcriptomes to study their heterogeneity in complex biological phenomena such as hematopoietic aging.

## HSPCs clustering reveals age-related differences

To test whether our dataset captured the expected age-related shifts in cell-cycle status, we projected phase scores onto the global UMAP. Seurat analysis showed that G1-phase cells clustered at the top of the manifold (Fig. 4A) and became progressively more abundant from young to old mice (Fig. 4B), consistent with the increased quiescence of aged HSCs. Overall purity was adequate for this analysis: 51.8% of sequenced cells carried the Sca1+Kit+Flt3−CD150+CD48− scGate label (Fig. 3B) and expressed canonical HSPC genes (Fig. 3E) plus markers of undifferentiated progenitors such as Procr and Notch2. Lineage-specific transcripts—Il7r and Cd79a for lymphoid cells, Elane and Mpl for myeloid cells—were minimal or absent (Supplementary Fig. 5A), confirming limited contamination. To resolve finer cell-cycle differences, we reclustered just the HSPC subset (Supplementary Fig. 4B), revealing 10 transcriptionally distinct clusters that form the basis for the detailed ageing analysis.

After successfully running a second post-clustering quality control check (Supplementary Fig. 4C), we observed that G1 phase cells dominated the UMAP representations (Fig. 4C), with an increasing proportion with age at the expense of the G2/M and S phases (Fig. 4D), showing the same trend throughout aging compared to that of the larger dataset (Fig. 4B). We then analyzed the gene expression signatures provided by Héuralt *et al*[35], focusing on the HSPC subset. We observed that most of these corresponded to non-primed gene expression states (Fig. 4E), specifically np1-4, interferon response (*ifn*), growth factor signaling (*tgf*), *rep*, *diff* (characterized by genes associated with progenitor restriction, such as *Cd34*), rep (enriched for DNA repair and replication genes) and *div* (associated with enrichment of the cell cycle KEGG pathway), as defined verbatim by Héuralt et al[35].

To further compare our dataset to that by Héuralt et al[35], we constructed a pseudotime trajectory using our larger dataset and selected the ifn cluster as the root, based on its highest proportion of HSPCs,[35] mirroring their Monocle analysis workflow[36]. As expected, the early pseudotime states aligned with non-primed signature clusters and branched toward more differentiated clusters located at the

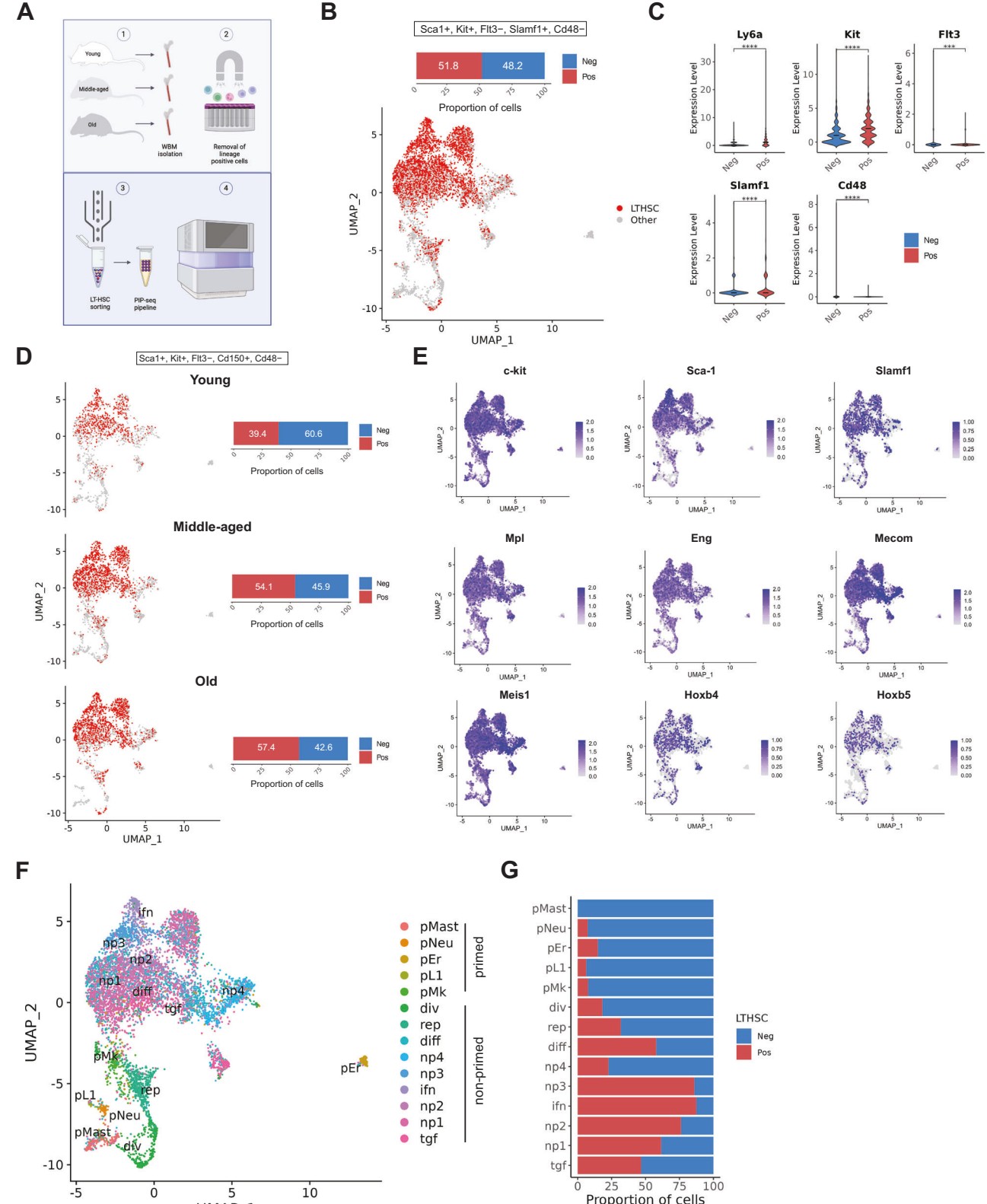

**Fig. 3 | PURE-Seq isolates murine long-term repopulating hematopoietic stem cells and enables single-cell sequencing via PIP-seq and analysis throughout aging. A** Schematic of the PURE-seq pipeline for sorting murine LT-HSCs from young, middle-aged, and old mice after depleting lineage-positive cells for scRNA-seq library preparation using PIP-seq and Illumina sequencing. **B** UMAP plot of all cells and scGate analysis of HSPCs. **C** Violin plots showing differences in the gene expression profiles of scGate-identified LTHSCs and non-LTHSCs. **** indicates *p*-value < 0.0001 from two-tailed unpaired Student's t-test. **D** UMAP plots showing the LTHSCs proportion across young, middle-aged, and old groups. **E** UMAP plots from the integrated dataset showing cells expressing key LT-HSC signature genes. **F** UMAP displaying identified cell populations in the integrated dataset, annotated according to Hérault et al[35]. **G** Proportion of LTHSCs by cell type.

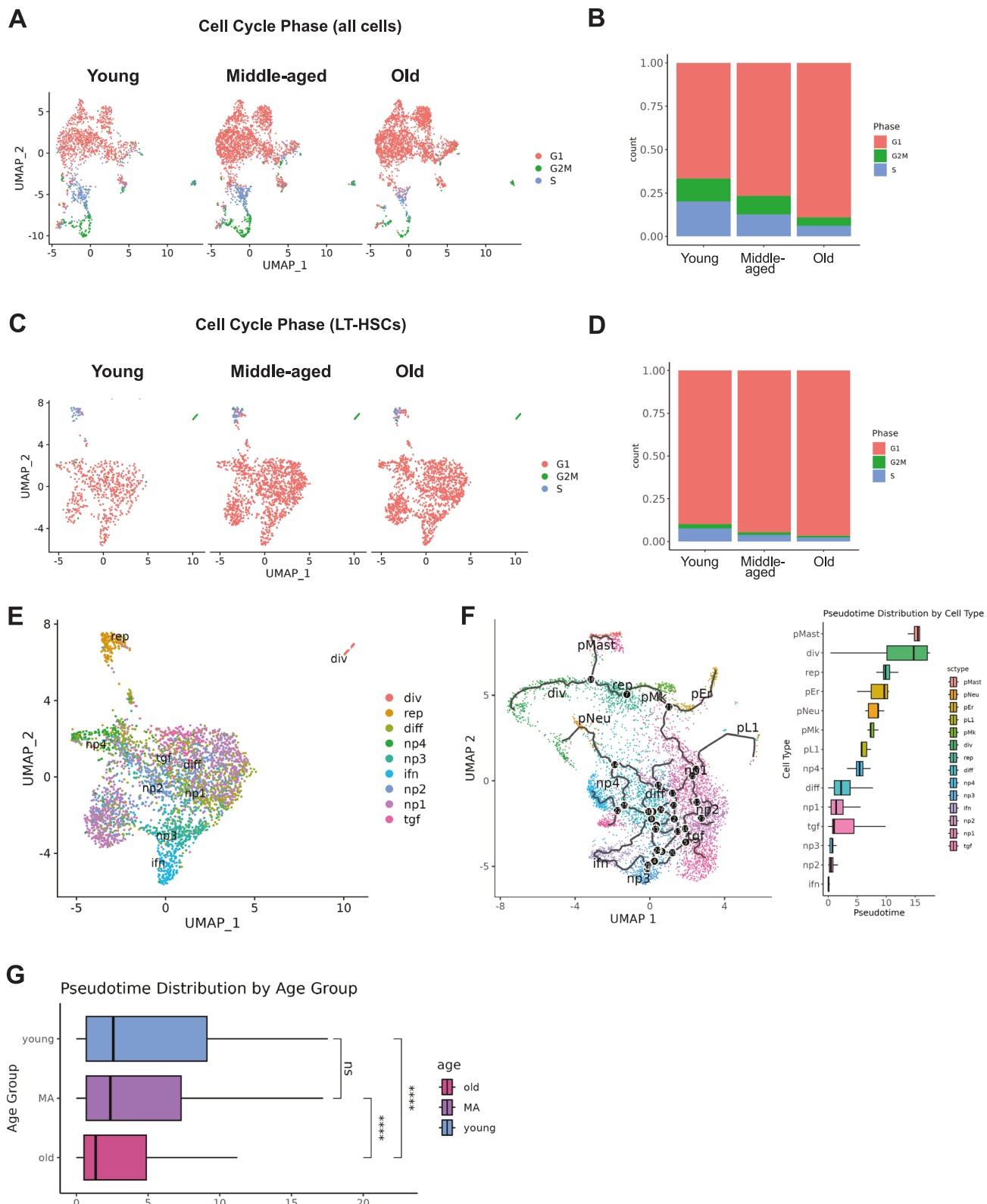

**Fig. 4 | scGATE marker-based purification, cell cycle analysis, and re-clustering of LT-HSCs. A** UMAP plot indicating the purity of LT-HSCs using scGate. **B** Analysis of cell cycle phases in the integrated UMAP plot. **C** UMAP plot of re-clustered LT-HSCs as per the scGate label. **D** Analysis of cell cycle phases in the re-clustered (purified) LT-HSC population. **E** UMAP plot of LT-HSCs labeled by cell types as annotated by Hérault et al[35]. **F** UMAP with pseudotime differentiation trajectory generated using Monocle 3 with all cells, using ifn cells as root. Boxplots (right) of pseudotime values are colored according to the cell type. **G** Boxplot of pseudotime values for young, middle-aged, and old cells. ns indicates no significance, and **** indicates *p*-value < 0.0001 from two-sided Wilcoxon rank sum test. Box plots: center line, median; box bounds, interquartile range (IQR; 25th–75th percentiles); whiskers, minima and maxima within 1.5 × IQR; outliers not shown.

distal end of the pseudotime trajectory (Fig. 4F). Notably, old cells showed a significant delay in pseudotime progression compared to their young and middle-aged counterparts, whereas no significant difference was observed between young and middle-aged cells (Fig. 4G). This suggests that the old sample was more enriched for non-primed subsets. These findings were in line with the results reported by Héuralt et al[35], who similarly observed a delay in pseudotime progression in aged HSPCs.

Additionally, both our overall dataset (Supplementary Fig. 3H) and the HSPC subset (Supplementary Fig. 5D) enabled the detection of age-related differences in cell numbers across all the Seurat clusters. Cross-comparison with the Héuralt et al. dataset[35] (Supplementary Fig. 4), along with similar results obtained from the pseudotime analysis (Fig. 4E–G), demonstrates that the PURE-seq pipeline can reproduce key findings while analyzing less than half the number of cells (6725 versus 15,000 cells). Moreover, it allows for the inclusion of an extra condition (the middle-aged group); this ability is especially valuable in sample scarcity scenarios where cell numbers are a constraint.

### Egr1 regulates HSPC aging transcriptional programs

Aging causes genetic and epigenetic changes that lead to a decline in HSPC function and self-renewal[37]. Recent studies have identified genes that may regulate hematopoietic aging, revealing differences in gene expression and aging biomarkers, as well as an inclination towards myeloid-biased hematopoiesis as early as middle age in mice[38,39]. In this context, single-cell transcriptomics has been useful in identifying crucial genes that could be targeted in potential hematopoietic rejuvenation strategies. To explore whether we could identify a relevant gene determining HSPC gene upregulation in aging from our dataset, we performed differential gene expression analysis and generated a bubble plot with top-downregulated or upregulated genes during HSPC aging (Fig. 5A). Although most differences laid in the expression of genes involved in fundamental cellular processes, including DNA synthesis (e.g., Rrm2b), autophagy (e.g., Vmp1), and transcription (e.g., Cnot6), we observed that there was an overall elevated expression of genes regulating the immune system and inflammatory responses with aging, as previously shown[38,40]. These genes included jun B proto-oncogene (Junb), suppressor of cytokine signaling 3 (Socs3), metal-lothionein (Mt1), immediate early response 2 (Ier2), Krüppel-like transcription factor 4 (Klf4), death-associated protein kinase 1 (Dapk1) and genes encoding for members of the S100 protein family (e.g., S100a6, S100a9). We also found that metabolic genes showed noteworthy differences, including the upregulation of genes implicated in lipid metabolism (e.g., Slc22a27), glycogenesis (e.g., Phkg1), and growth factor signaling, such as the early growth receptor 2 (Egr2) and 3 (Egr3), and the expression of Egr1 (Fig. 5A).

Next, we performed ShinyGO Pathway Analysis[41] to identify significantly enriched cellular pathways in aged HSPC in an unbiased manner. We utilized the complete list of upregulated genes in old HSPCs compared to their middle-aged and young counterparts, respectively. The gene ontology category ribosome was the most significantly enriched gene set, which was an expected finding given the known altered upregulation in ribosomal gene transcription with hematopoietic aging, from which others have inferred that old HSPCs may be aberrantly activated through ribosomal biogenesis despite cycling less than younger cells[42]. The rest of the enriched pathways were mainly metabolism-related or linked to the pathogenesis of age-related diseases, such as cardiovascular or degenerative disorders (Fig. 5B). Using a web-based transcription factor (TF) enrichment analysis tool, ChEA3[43], which predicts the most probable TF network underlying gene expression changes based on existing epigenetic and transcriptomic datasets, we identified EGR1 as a core node of the resulting TF network driving the transcriptional upregulation in old HSPCs (Fig. 5C), with EGR1 emerging as the top-ranked hit (Fig. 5D). As

an orthogonal validation, we performed ChEA3 analysis using a DEG list previously generated by Young et al. (2022) from sorted Lin⁻Sca1⁺Kit⁺EPCR⁺Flt3⁻CD150⁺CD48⁻CD41⁺ HSCs—representing LT-HSCs—from young and old mice. EGR1 again appeared as a central component of the predicted TF network (Supplementary Fig. 6A), ranking within the top 20 hits (Supplementary Fig. 6B). In parallel, we conducted ChEA3 analyses using upregulated genes from the old vs. young and old vs. middle-aged comparisons. EGR1 was identified in the resulting TF networks from both analyses (Supplementary Fig. 6C). These findings suggest that Egr1 is involved in transcriptional changes across the entire aging spectrum, including within advanced age (i.e., from middle-aged to old), in murine HSCs.

UMAP analysis revealed wide distribution in the HSPC subset UMAP (Supplementary Fig. 6D). However, although the expression of Egr1 was not restricted to middle-aged and old HSPCs (Fig. 5E), its expression level notably increased in middle age, as observed in the bubble plot (Fig. 5A). Indeed, the expression level of Egr1 was found to be statistically significant when comparing young versus middle-aged, young versus old or middle-aged versus old HSPCs (Fig. 5E). These results suggest that the upregulation of Egr1 in middle age might be responsible for a subsequent gene program upregulation promoting murine HSC aging, with widespread Egr1 constitutive expression in old age to maintain it. We also sought to examine the relationship between cell cycle status and Egr1 expression in HSPCs during aging, given that Min et al[44] originally described a role for EGR1 in promoting HSC quiescence at the expense of cell cycle progression into the G2/M and S phases. Consistent with this, we observed that higher Egr1 expression within this HSC subset was associated with a lower proportion of cells in G2/M and S phases (Fig. 5G, Supplementary Fig. 6E). Furthermore, Gene Set Enrichment Analysis (GSEA) revealed that Egr1-high cells showed upregulation of inflammatory gene pathways across all age groups. Notably, these inflammatory signatures closely overlapped with those observed in the GSEA comparing young and old HSPCs, regardless of Egr1 expression.

Overall, these data demonstrate that the PURE-seq pipeline can enrich and sequence rare cell populations from primary samples, such as murine HSPCs, to generate high-quality single-cell transcriptomes and, in so doing, give valuable insights into complex biological processes, as it is hematopoietic aging. Compared with existing pipelines, PURE-seq offers a user-friendly solution requiring significantly fewer cells while delivering comparable quality data, which is suitable for biological analyses of rare cell populations.

## Discussion

PURE-seq enables the recovery and sequencing of rare cells from mixed populations by integrating two commercially available platforms: FACS and PIP-seq. PIP-seq allows cell barcoding within standard Eppendorf tubes—commonly used vessels for cell recovery in FACS protocols. This direct integration eliminates cell transfer steps, significantly reducing cell loss and enabling the reliable capture and sequencing of ultra-rare cells.

Our study demonstrates that PURE-seq can enrich and analyze murine HSPCs comparably to current methods, such as 10X Genomics, even when using only half of the input cells. This approach is cost-effective, compatible with readily available commercial systems, and opens doors for proteomic analysis, including technologies like CyTOF[45] and Abseq[46], as well as multiomics through CITE-seq[47]. PURE-seq has the potential to significantly contribute to genomic and proteomic investigations, particularly those focusing on ultra-rare cell populations that can be enriched using flow cytometry. Furthermore, PIP-seq can be combined with antibody-based cell hashing[7]. Although we did not perform hashing in this study, it can be used to further increase the number of cells and conditions processed in the PIP-seq pipeline.

PURE-seq offers potential for studying CTCs, which remain challenging to profile due to their rarity[1,2,14,48]. Our melanoma CTC

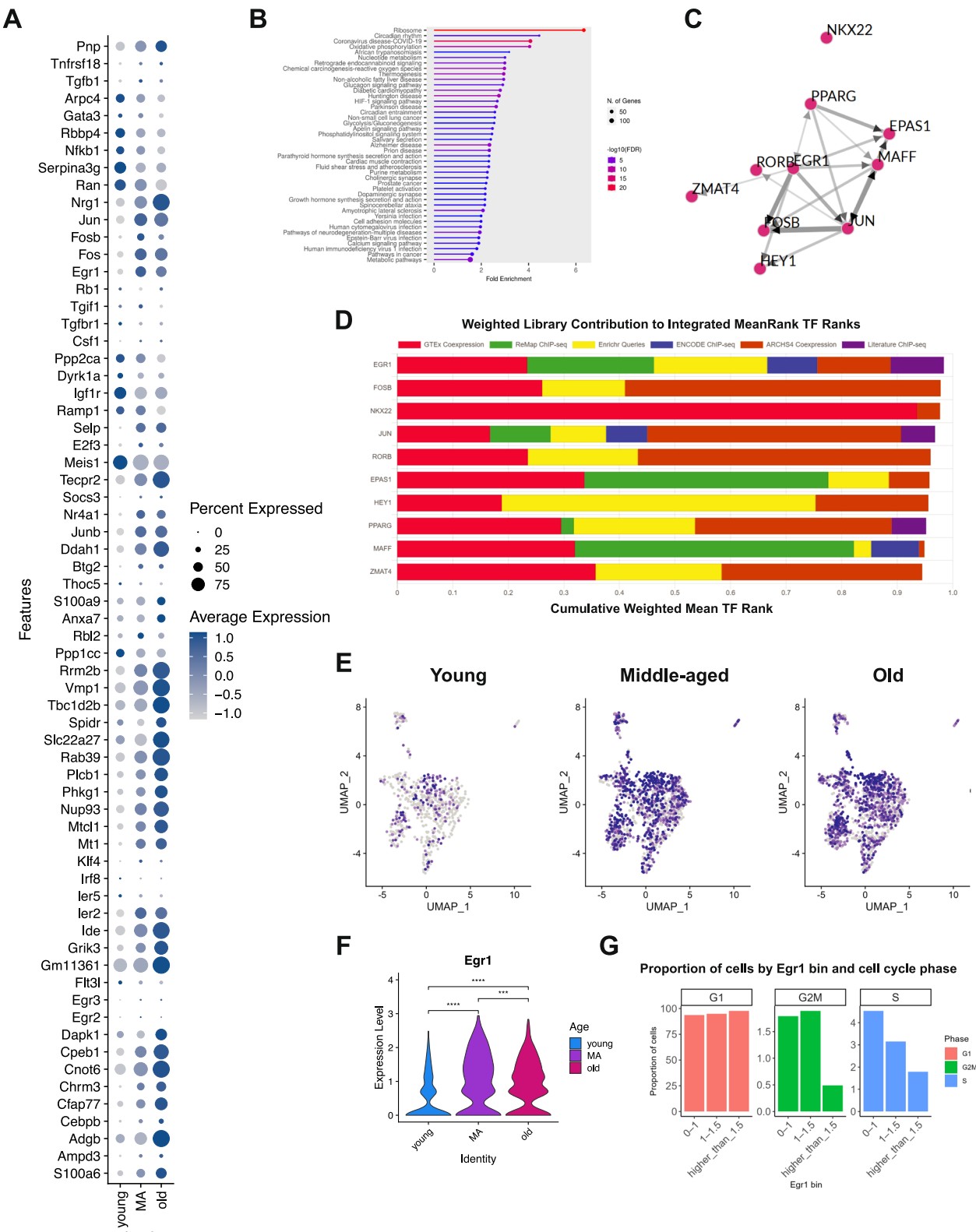

**Fig. 5 | Identifying *Egr1* as a potential master regulator gene in the gene expression signature of aged murine long-term repopulating hematopoietic stem cells. A** Bubble plot of the top downregulated/upregulated gene signature of old LT-HSCs compared to their young and middle-aged counterparts. The color of the spheres indicates the average gene expression, and their size represents the percentage of cells expressing each gene. **B** The ShinyGO Pathway Analysis[41] illustrates the top-enriched pathways in aged LT-HSCs compared to their young and middle-aged counterparts. The circle size represents the number of differentially expressed genes classified into one specific pathway category. **C** Transcription

factor network derived from the top upregulated genes in aged LT-HSCs based on the ChEA3 analysis[44]. **D** Weighted library contributions to the integrated mean-rank of transcription factors from ChEA3 analysis of upregulated genes in aged HSPCs. **E** UMAP plots showing *Egr1*-expressing cells in young, middle-aged, and old LT-HSC samples. **F** Violin plots showing *Egr1* expression in young, middle-aged, and old LT-HSC samples; *p*-values from two-tailed unpaired Student's *t*-test, indicating a *p*-value less than 0.0001 (****) or no significance (ns). **G** Proportion of HSPCs by Egr1 expression bin and cell cycle phase.

analysis demonstrates PURE-seq's capacity to both identify and sequence CTCs in peripheral blood, including CTC populations not amenable to capture by traditional enrichment methods[1,2,14,48] such as EpCAM, which remains the only FDA-approved approach for CTC capture[1,14] but misses CTCs lacking canonical surface markers. In contrast, the throughput and workflow flexibility of PURE-seq enabled negative enrichment, broadening the CTC detection spectrum by allowing recovery of CTCs that may not express EpCAM. Our PURE-seq analysis uncovered two CTC subpopulations with divergent gene expression signatures highlighted by KRAS gene set upregulation and CNV profiles. Of note, while the inferCNV analysis supported the malignant identity of CTCs, RNA-based CNV inference is inherently noisy, and the alterations in reference cells likely reflect data sparsity rather than true genomic changes. Future integration of scRNA-seq or DNA-seq data of the primary melanoma could improve CNV accuracy and confidence. Although the sample size in our pilot study precludes any robust inference regarding the translational significance of these two CTC subpopulations, the observed CTC heterogeneity could be clinically actionable, given the disparate molecular signatures observed at the DNA and RNA levels in these two CTC clusters. Notably, the PURE-seq workflow is fully compatible with routine clinical laboratory infrastructure and standard FACS platforms, distinguishing it from specialized microfluidic CTC enrichment systems that are often challenging to implement at scale[49,50]. In practice, PURE-seq's accessibility and the robustness of commercial PIP-seq kits make it feasible for conducting larger, longitudinal studies across diverse clinical settings. Using the yield sorting precision mode, we can leverage high-throughput single-cell sequencing downstream of FACS isolation to recover single CTC transcriptomes, even when mixed with non-CTCs. Although this approach may increase false positives, the high-resolution single-cell data can still identify the relevant CTCs through transcriptomic signatures. Thus, PURE-seq holds significant promise for translational applications, and future prospective analysis of CTCs will be important to determine whether these signatures may be useful both as a biomarker of disease progression and to understand mechanisms of treatment resistance that could inform future therapy selection[49,50].

We further applied PURE-seq to investigate hematopoietic aging in murine HSPCs. While prior studies have shed some light on HSC transcriptional circuitry[44,48], the role of *Egr1* in murine HSC aging has not yet been fully elucidated. Recent studies involving scRNA-seq and bulk RNA sequencing have indicated increased *EGR1* expression in aged human HSPCs[15,16]. EGR1 may regulate HSPC quiescence, proliferation, and localization, making it crucial in determining their function and fate. It has been suggested that reducing EGR1 expression may reactivate aged HSPCs, potentially improving their function and offering a target for hematopoietic rejuvenation strategies[17].

Using PURE-seq in combination with the ChEA3-derived prediction[43], we identified *Egr1* as a putative master regulator of HSC aging in mice, supporting previous evidence in the field and providing a foundation for future genomic, epigenomic, and mechanistic studies. The role of *Egr1* as a TF relevant to murine HSC function was first described by Min et al. (2008)[44], who identified *Egr1* as essential for maintaining HSC quiescence and retention within the stem cell niche. More recently, Stoddart et al. (2022) demonstrated that *Egr1* regulates genes involved in differentiation, inflammation, and DNA damage responses[51]. They also showed that *Egr1* haploinsufficiency provides a competitive advantage to HSPCs subjected to stress from serial bone marrow transplantation, consistent with its previously described role as a haploinsufficient myeloid leukemia gene in the context of N-ethylnitrosourea-induced mutagenesis[52]. Interestingly, *Egr1* expression increases in middle age in our dataset, suggesting a potential role as an early master regulator of HSC aging. This finding supports the idea that

hematopoietic aging begins in middle age[38] and may point towards a functional link between age-associated hematopoietic DEG and myeloid cell transformation. It also underscores the need for future studies investigating how the temporal dynamics of gene expression contribute to HSC aging. Yet, it remains essential to functionally characterize HSPC subpopulations in relation to *Egr1* expression, as observed changes may reflect shifts in the proportion of HSCs differentiating into progenitors or a more lineage-restricted progeny expressing *Egr1*. Nonetheless, we hypothesize that these changes probably originate within the HSC compartment, based on the observed negative correlation between *Egr1* expression and the proportion of cells in the G2/M and S phases of the cell cycle. This relationship suggests that *Egr1*<sup>high</sup> cells are less proliferative, a characteristic more consistent with bona fide HSCs. However, this interpretation remains speculative and warrants functional validation to determine the precise role of *Egr1* in HSC aging.

Although our identification of *Egr1* in HSC aging using PURE-seq is descriptive and based on the CheAR-3 prediction within the context of our dataset, it illustrates the power of the PURE-seq pipeline to identify candidate genes involved in complex and heterogeneous biological processes. Notably, PURE-seq achieves this using a user-friendly workflow and ≤2500 cells per sample—significantly fewer than traditional, more labor-intensive methods—making it particularly advantageous for studying rare cell populations. Furthermore, comparison with the dataset from Hérault et al[35] validates the biological and technical comparability of PURE-seq to 10X Genomics, demonstrating its ability to yield high-quality data and meaningful biological insights for research.

Recent advances in single-cell technologies have introduced innovative methods to enrich rare-cell populations based on transcriptomic features. PERFF-seq[53] uses RNA FISH to target specific transcripts, enabling the profiling of fixed nuclei, including those from FFPE samples. Incorporating RNA FISH-based enrichment from PERFF-seq into PURE-seq could expand its applicability to challenging fixed tissues. Similarly, FIND-seq's nucleic acid cytometry using microfluidic droplets offers another potential avenue for adaptation[54]. By integrating FIND-seq's enrichment principle into PURE-seq's microfluidics-free workflow, it may be possible to achieve single-molecule sensitive nucleic acid-based cell enrichment. Together, these approaches demonstrate how PURE-seq could evolve to address a broader range of sample types and rare-cell applications and inform biological studies of a spectrum of important research questions.

## Methods

### Ethics

This research complies with all relevant ethical regulations. Human blood sample collection and analysis were approved by the UCSF Institutional Review Board (protocol 138510), and all patients provided informed consent to contribute biospecimens for the molecular study of resistance or response to therapy in melanoma. No compensation was received for participating in this study. All animal experiments were performed in accordance with institutional guidelines and approved by the Institutional Animal Care and Use Committee of Memorial Sloan Kettering Cancer Center (protocol #07-10-016).

### PURE-seq workflow

PURE-seq combines Fluorescence-activated cell sorting (FACS) and Particle-templated instant partition sequencing (PIP-seq) in an integrated workflow. For the mouse-human mixing experiments described herein, the BD FACS Aria system was used for sorting, and Sweetspot was turned on to ensure a stable stream during the sorting. The cooling unit was set to 4 °C to keep the collection unit with the PIP-seq reaction tube cold throughout the sort. A 0.5 mL tube adapter (Cole-Parmer, EW-17414-73) was inserted into the Aria 1.5 mL collection tube holder to hold the PIP-seq T2 tube. Then, we fine-tuned cell sort stream

alignment by using an empty 0.5 mL Eppendorf tube to make sure the test sort droplet was located at the center of the lid when the lid was closed and at the center of the tube bottom when the lid was open. For quality control of each sorting session, we quantified the sorting recovery rate by sorting 100 Calcein-labeled cells into a 0.5 mL Eppendorf tube pre-loaded with 10 μL media and counted the number of cells collected under the microscope. The recovery rate is calculated as # Target cells counted under the microscope / # Target cells reported to have been sorted by the instrument. To optimize cell viability and capture efficiency, we capped the total sorting duration to 60 min and the total sorted volume to 5 μL (2500 drops with 85 μM nozzle). Based on BD FACS Aria's instrument specifications, we limited the flow rate to no more than 8 kHz to minimize shear stress during sheath flow focusing (i.e., 8000 events per second with 85 μm nozzle). Once the sorting was complete, the PIP-seq T2 tube was unloaded to proceed to the standard PIP-seq protocol from Cell Capture and Lysis after the cell loading step to the preparation of the scRNA-seq library.

## Mouse-human mixing experiment

Human HEK 293 T and mouse NIH 3T3 cells (ATCC) were cultured in Dulbecco's modified Eagle's medium (DMEM, Thermo Fisher, 11995073) supplemented with 10% fetal bovine serum (FBS; Gibco, 10082147) and 1× Antibiotic-Antimycotic (Gibco, 15240062) at 37 °C and 5% $CO_2$. Cells were treated with 0.05% Trypsin-EDTA with Phenol red (Gibco, 25200114) for 3 min, quenched with growth medium, and centrifuged for 3 min at 300 $g$. The supernatant was removed, and the cells were resuspended in 1X DPBS without calcium or magnesium. Fresh-frozen human peripheral blood mononuclear cells (PBMCs) were obtained from STEMCELL Technologies. DMEM with 10% FBS was warmed up to 37 °C, and the frozen PBMCs were thawed by adding 1 mL of warm media on top of the frozen cells and immediately transferring the media to a 15-mL conical. This process was repeated until all PBMCs were thawed and transferred. Cells were centrifuged for 3 min at 300 $g$ and resuspended in 1X DPBS. For the $10^{-3}$, $10^{-4}$, and $10^{-5}$ target cell fraction samples, human HEK 293 T cells were the target population mixed with a mouse NIH 3T3 cells background population. For the $10^{-6}$ target cell fraction sample, mouse NIH 3T3 cells were the target population mixed with the human PBMCs background population. The target population was treated with 1 μg/mL Calcein Red-Orange (Invitrogen, C34851), and the background population was treated with 1 μg/mL Calcein Green (Invitrogen, C34852) for 15 min at 37 °C, followed by washing and dilution to the final concentration in 1× DPBS with 0.1% BSA. The viability and cell concentration were evaluated by an automated cell counter (Bio-Rad, TC20) after adding Trypan Blue (Gibco, 15250061). The mixed cell suspension was filtered through a 40 μm cell strainer (Flowmi, BAH136800040) and processed through the PURE-seq workflow described above to enrich for Calcein Red-Orange labeled cells. For this experiment, we selected the yield sorting mode to ensure as many rare cells were sorted, set the flow rate to 8 kHz, and restricted the sorting duration to 60 min or if the total sorted volume of 5 μL (2500 drops with 85 μm nozzle) was reached. In the sequenced libraries, cell transcriptomes were aligned to the human or mouse genome to quantify for PURE-seq sensitivity and specificity.

## Sorting precision modes experiment

Calcein Red-Orange labeled human HEK 293 T cells and Calcein Green labeled mouse NIH 3T3 cells were mixed at a ratio of 1:1000. The mixed sample volume was controlled at 1 mL. Each sample was processed through the PURE-seq workflow described above using yield or single-cell sorting precision mode until depletion of the sample.

## Optimization of sorting parameters

To maximize PURE-seq performance across cell types, we recommend the following guidelines:

1. **Cell preparation and buffer composition**

   - Cells should be verified for 90% viability and filtered before sorting
   - Cell concentration should be ≤10 million/mL for robust cell lines, ≤8 million/mL for primary cells, and ≤5 million/mL for adherent cells
   - Add 0.1–0.5% BSA or 2–5% FBS to reduce adhesion losses
   - For stress-prone cells: supplement with 1 mM EDTA to prevent clumping or use pre-made FACS buffer for specific cell types

2. **Sorting precision modes**

   - Use yield mode for rare cells (< 1:10,000) or fragile primary cells to maximize recovery
   - Apply single-cell mode for pre-enriched populations (> 1:10,000) or robust cell lines to prioritize purity

3. **Nozzle selection**

   - 85 μm nozzle for small cells (< 20 μm)
   - 100 μm nozzle for larger cells (> 20 μm) or fragile cells
   - Avoid 70 μm nozzles for primary cells to minimize shear stress

4. **Flow sate**

   - ≤6 kHz for primary cells
   - ≤8 kHz for cell lines

5. **PURE-seq reaction tubes**

   - Select appropriate reaction tubes (T2, T20, T100) based on desired sequenced cell number
   - Set the cooling unit to 4 °C to keep the PURE-seq reaction tubes cold throughout the sort
   - Confirm stream alignment to PURE-seq reaction tubes using test sort on the reaction tube cap or HRP-TMB droplet alignment tests

6. **Duration/volume limits**

   - Cap sorting for each PURE-seq reaction tube at 60 min or input collected volume (i.e., 5 μL for T2 tubes)
   - Use the Sort Droplet Volume Calculator for the number of drops sorted allowed for the volume limits, with different nozzles or reaction tubes selected

## Patient PBMC processing for CTC enrichment

Peripheral blood (8 mL/patient) was collected from metastatic melanoma patients in EDTA tubes (BD Vacutainer, 367861) and processed for PBMCs isolation with Histopaque-1077 (Sigma-Aldrich) under IRB 22-37134. Then, the cells were washed once with Cell Staining Buffer (Biolegend, 420201) to be stained and sorted immediately. The cells were then stained with human TruStain FcX (Biolegend, 422301), APC anti-human CD45 (Biolegend, 304011), and Calcein AM (Biolegend, 425201). After staining, the cells were washed once with Cell Staining Buffer (Biolegend, 420201) and filtered through a 40 μm cell strainer (Flowmi, BAH136800040). We sorted the cells into PURE-seq T2 tubes using a BD FACS Aria III (100 μm nozzle, 4 °C) following the protocol described in the Pure-seq workflow section and using the yield sorting precision mode to maximize the recovery level. For each PBMC sample, two PURE-seq T2 tubes were prepared for sequencing: 1) 2000 live, CD45+ cells were first sorted for control; 2) live, CD45- cells were sorted for CTC enrichment until depletion of the sample.

Libraries were sequenced on an Illumina MiSeq, targeting >50,000 reads per cell, at the UCSF Center for Advanced Technology.

Library demultiplexing, read alignment, identification of empty droplets, and UMI quantification were performed using PIPseeker 1.0.0 (Fluent BioSciences) with default parameters.

### scRNA-seq data analysis for patient PBMCs

PIPseeker-generated filtered feature matrices were imported into a Seurat object and analyzed with Seurat v4.3.0. For quality control, data were filtered on a per-sample basis to remove outliers in gene count, UMI count, mitochondrial genes, and ribosomal genes. The individual count matrices were normalized by SCTransform v2, and Scanorama (https://github.com/brianhie/scanorama) was used to perform data integration across datasets. Tumor versus non-tumor cell designation was performed through a combination of manual marker gene inspection, Jensen Diseases database enrichment (Enrichr), and inferCNV (https://github.com/broadinstitute/inferCNV) of the CD45- clusters using the CD45+ sorted hematopoietic cells as reference. GSEA analysis was performed using fgsea (v1.27.1) and msigdbr (v7.5.1).

For comparison with healthy PBMCs, a publicly available scRNA-seq dataset was downloaded from the 10X Genomics database (https://www.10xgenomics.com/datasets/3-k-pbm-cs-from-a-healthy-donor-1-standard-1-1-0) and integrated with the PURE-seq datasets above for a combined analysis.

### Experimental animals

The study with primary mice was performed in accordance with institutional guidelines established by Memorial Sloan Kettering Cancer Center under the Institutional Animal Care and Use Committee-approved animal protocol (#0710016) and the Guide for the Care and Use of Laboratory Animals (National Academy of Sciences 1996). Mice were maintained under specific pathogen-free conditions in a controlled environment that maintained a 12-h light-dark cycle, and food and water were provided *ad libitum*. The following mice were used: young (2–3 months old), middle-aged (12–14 months old), and old (18–20 months old) female C57BL/6 mice. Young mice were purchased from the Jackson Laboratories and either used when young or aged in-house until middle age. Old mice were obtained from the National Institute of Aging (NIA) and acclimatized for at least 2 weeks at our facility before use. Mice were healthy, had intact immune systems, and had not undergone any prior procedures before euthanasia. For each cohort, 4–6 mice were used to make 2–3 pooled age-matched bone marrow (BM) samples per group prior to sorting.

### Mouse bone marrow harvesting and sample processing for sorting

Mice were humanely euthanized using $CO_2$. BM cells from their limb bones were isolated and resuspended in FACS buffer (PBS + 2% FBS) by centrifugation at $8000 \times g$ for 1 min. After removing red blood cells (RBC) with a commercial lysis buffer (BioLegend, 420302), diluted to 1X with distilled water, single-cell suspensions were depleted of hematopoietic cells committed to a specific lineage using a Lineage Cell Depletion Kit (EasySep, StemCell Technologies, Inc., 19856 A), according to the manufacturer's instructions. To label LT-HSC cells, the following fluorophore-conjugated antibodies were used at the indicated dilutions: CD117 (c-Kit) BV785 (clone 2B8, BioLegend; 1:200 dilution), Ly-6A/E (Sca-1) PE/Cy7 (clone D7, BioLegend; 1:1000 dilution), CD48 PerCP/Cy5.5 (clone HM48-1, BioLegend; 1:100 dilution) and CD150 (SLAM) APC (clone TC15-12F12.2, BioLegend; 1:50 dilution). After adding the rat serum and isolation cocktail of the Lineage Cell Depletion Kit, the LT-HSC-labeling antibodies were also added for a 30-minute-long incubation in the dark at 4 °C. Following the removal of lineage-positive cells, samples were spun down in FACS buffer and subsequently resuspended in 200–300 µL of FACS buffer containing DAPI at a final concentration

of 1 µg/mL. Cells from 2/3 age-matched mice were combined to generate each pool sample, with a total of 2 replicates for the young condition and 3 replicates for the middle-aged and old conditions, respectively (total n = 10 mice). Before sorting, we also performed the Rmax method to calculate the maximum recovery of the sample sort and a sorting test with horseradish peroxidase (HRP) using a 0.5 mL collection tube containing a drop of a 3,3′,5,5′-tetramethylbenzidine (TMB), which turned blue if the HRP fell directly into the tube center. Leveraging this HRP-TMB reaction, we ensured that the instrument alignment was correct so that the sample was sorted straight into the PIP-seq T2 reaction. All the mouse primary samples were sorted using a Spectrally Enabled (SE) five-laser BD FACSymphony™ S6, following the protocol described in the Pure-seq workflow section and using the single-cell sorting precision mode to maximize the purity level.

### scRNA-seq library preparation and sequencing

Single cells were processed for scRNA-seq using the PIP-seq T2 3′ Single Cell RNA kit according to the manufacturer's protocol (Fluent Biosciences, FB0001026, 20210908 version). cDNA and final library DNA quality were confirmed using a 2100 Bioanalyzer Instrument (Agilent Technologies). Libraries were pooled at equimolar ratios and sequenced on an Illumina NovaSeq 6000 S4 platform at PE100 (200 cycles), targeting >50,000 reads per cell. Library demultiplexing, read alignment, identification of empty droplets, and UMI quantification were performed with PIPseeker 1.0.0 (Fluent BioSciences) with default parameters.

### scRNA-seq data analysis for mouse HSPCs

Filtered feature matrices were imported into Seurat, and all downstream analyses were performed using Seurat v4.3.0[55]. For quality control, data were filtered to remove outliers in gene count, UMI count, mitochondrial genes, and ribosomal genes. The 8 samples (young 1–2, middle-aged 1–3, and old 1–3) were normalized by SCTransform and then integrated by the Seurat integration workflow[55]. A PCA was estimated, and clustering was performed on 20 principal component dimensions (selected by visual analysis of an Elbowplot). A uniform manifold approximation and projection (UMAP) embedding was calculated using the selected 20 principal components as input. Cell cycle was not regressed. [28] As LT-HSCs were of interest in this study, scGate R package[28], a marker-based purification algorithm, was applied to annotate the cells based on each cell's gene expression and the defined gate (Ly6a + , Kit + , Flt3-, Slamf1 + , Cd48−). The target cell population was then extracted, re-embedded, and re-clustered, followed by a second post-clustering quality control step for further in-depth analysis. From the identified clusters, differential gene expression analysis was conducted using the Seurat function FindAllMarkers to identify genes that were significantly up/downregulated in specific cell clusters compared to others.

For direct comparison with a 10X Genomics dataset, the published 10X scRNA-seq dataset by Héuralt et al[35] was integrated with the PURE-seq dataset for a combined analysis. Importantly, the lineage depletion and cell-gating strategy described in that study was replicated prior to sequencing of the PURE-seq samples. After Seurat integration, clustering, and differential gene expression analysis, marker genes were compared between the two methods and gene expression profile correlations were computed for each cluster. The cells were then annotated using the ScType automated cell type classification[56] with a custom marker panel defined by Héuralt et al[35], which comprises 15 HSC subtypes, including 6 primed types (pMast, pNeu, pEr, pL2, pL1, pMk) and 9 non-primed types (div, rep, diff, np4, np3, ifn, np2, np1, tgf). Finally, a pseudotime trajectory was inferred across all PURE-seq HSCs using Monocle 3 to visualize the differentiation trajectory. The root state was defined as the cell subtype

with the highest proportion of LT-HSCs, in accordance with the approach of Héuralt et al. For gene GSEA, we used the GSEA software version 4.3.3 and analyzed the following gene sets: (I) Reactome_Interferon_Gamma_Signaling, (II) Reactome_Cytokine_Signaling_in_Immune_System, (III) Hallmark_TNFA_Signaling_via_NFKB, (IV) Hallmark_MTORC1_Signaling, (V) Hallmark_Inflammatory_Response, (VI) Hallmark_IL6_JAK_STAT3_Signaling, (VII) Hallmark_Hypoxia, and (VIII) Hallmark_Cholesterol_Homeostasis. These datasets were selected to closely match those used by Young et al. (2022)[38], and were obtained from the GSEA website (https://www.gsea-msigdb.org/gsea/index.jsp). Results were plotted based on enrichment scores. To predict the transcriptional regulatory network associated with gene upregulation in aging Sca1$^+$Kit$^+$Flt3$^-$CD150$^+$CD48$^-$ HSCs, we used ChIP-X Enrichment Analysis 3 (ChEA3), a web-based transcription factor enrichment analysis tool (https://maayanlab.cloud/chea3/). No mouse sample was excluded from any of these analyses.

### Statistics & reproducibility

No statistical method was used to predetermine sample size. The sample sizes used were consistent with those in similar single-cell transcriptomic studies and were sufficient to detect clear biological effects. Statistical analyses were performed in R (v3.6.1) unless otherwise specified. Differences between groups in single-cell RNA-seq data were assessed using two-tailed unpaired Student's *t*-test and two-sided Wilcoxon rank-sum tests. All computational analyses are fully reproducible using the code and parameters described in the Methods.

### Reporting summary

Further information on research design is available in the Nature Portfolio Reporting Summary linked to this article.

## Data availability

Sequencing data were deposited into the NCBI Gene Expression Omnibus under GSE273803.

## Code availability

The open-source software, tools, and packages used for data analysis in this study, as well as the version of each program, were R (v3.6.1), PIPseeker (v1.0.0), Seurat R package (v4.3.0), scGate R package (v1.6), ScType R package (v1.0), and SingleR R package (v1.0)[57]. No custom software, tools, or packages were used.

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

## Acknowledgments

This work was supported by grants R01AI149699 and R01NS130876. I.F-M. was supported by a postgraduate fellowship from the La Caixa Foundation, a Momentum Fellowship from the Mark Foundation for Cancer Research, a Scholarship of Excellence Rafael del Pino, and an NCI F99 award (CA284253-01). R.L.B. was supported by a Damon Runyon-Sohn Fellowship and the NCI (K99CA248460). R.L.L. was sup-ported by a Memorial Sloan Kettering Cancer Center Support Grant/ Core Grant P30 CA088748, an R35 grant from the National Institute of Cancer (CA197594), and a collaborative U01 Research Project grant from the National Institute of Aging—the U01 grant was jointly received with the laboratory of Jennifer Trowbridge at the Jackson Laboratories (U01AG077925; 210374-0622-02). We are grateful to members of the Abate laboratory for helpful discussions. We thank Eric Chow and the staff of the UCSF Center for Advanced Technology for their technical support, the members of the Flow Cytometry and the Integrated Genomics Operation (IGO) cores at Memorial Sloan Kettering Cancer Center for their advice and technical help, Kristina Fontanez, Autumn Cholger and Bob Wikle from Fluent BioSciences for their advice and support.

## Author contributions

S.P., K.C., and A.R.A. designed the study; S.P. and K.C. optimized the PURE-seq workflow; I.F.-M. designed and performed the experiments for all mouse studies; S.P., A.M.H, K.K.T, A.I.D., and H.N.V. conducted the CTC study; I.F.-M, K.C., and S.P. analyzed scRNA-seq data; S.V.H. pro-vided bioinformatic and data curation support; M.G.W. assisted with mouse dissections and sample processing; R.L.B. provided input on data visualization; S.P. and I.F.-M. wrote the manuscript; A.R.A. and R.L.L. supervised the work and revised the manuscript; all authors read, reviewed, and approved the manuscript.

## Competing interests

R.L.L. is on the Supervisory board of Qiagen (compensation/equity), a co-founder/board member at Ajax (equity), and is a scientific advisor to Mission Bio, Syndax, Scorpion, Zentalis, Auron, Prelude, and C4 Ther-apeutics; for each of these entities, he receives equity/compensation. He has received research support from the Cure Breast Cancer Foun-dation, Calico, Zentalis and Ajax, and has consulted for Jubilant, Gold-man Sachs, Incyte, AstraZeneca and Janssen. The remaining authors declare no competing interests.
