## [Transparent Peer Review file · Nature Communications]

PURE-seq Integrates FACS and PIP-seq for Single-Cell Genomics of Ultra-rare Cells

Corresponding Author: Professor Adam Abate

Version 0:

Reviewer comments:

Reviewer #1

(Remarks to the Author)

In this manuscript, Pan et al., introduce and benchmark PURE-seq, a single-cell transcriptomics method with selective enrichment of cells prior to sequencing allowing for an increased capture of rare cells. The assay relies on the direct loading via FACS-sorting into PIP-seq reactions, thereby reducing cell loss. PURE-seq uses standard laboratory equipment which makes it more readily accessible. Finally, the authors apply PURE-seq to characterize LT-HSCs in murine aging and identify *Egr1* as a potential regulator in this context.

Overall, the manuscript is very well written, and the quality of the experiments and data is certainly very high. This Reviewer has the following comments:

Major comments

1. Enriching rare cells and directly loading them into droplets to minimize loss is a very useful tool for the community and a helpful addition to the PIP-seq workflow. However, from a technical perspective, the combination of steps is quite logical, rather than presenting an innovative new protocol. In principle, a similar level of enrichment could be achieved with most other downstream single-cell genomic workflows. While the level of cell loss may be variable across platforms (e.g., including 10x), this has not been formally shown and is likely also only to be increasingly problematic with the increasing rarity of the cell population of interest.

As such, the authors could focus more on the technology providing hands-on guidelines such as how to fine-tune sorting settings for PURE-seq. For example, I assume optimal sorting settings are different for different cell types and the cell loading concentration could be quite variable depending on the prevalence of the cells of interest. Also, for rare cells more input material and long sorting times are necessary and therefore cells would have to remain viable in the FACS and downstream buffers. Therefore, I think it would be valuable to showcase the approach for another cell type/tissue such as, for example, circulating tumor cells (as the authors pointed out in the discussion), and discuss PURE-seq workflow parameters.

2. The biological novelty of the described findings appears rather limited and primarily serves the purpose of a showcase of the PURE-seq workflow. The observations regarding the increase in numbers of LT-HSCs, their gradual loss of self-renewal with aging as well as the identification of *EGR1* as key transcription factor do not seem very striking, and additional quantifications should be shown. For example, the differences in percentage in flow and sequencing are rather subtle and are only shown for three data points with relatively low cell numbers and also without assessing variability between individual mice (e.g., Line 191: "percentage of LT-HSCs increased with age (Supplementary Figures 3A, B)"; Line 194: "(UMAP) plots that showed a higher number of hematopoietic cells [...] in the middle-aged and old samples compared to their young counterparts (Figure 3C)"; Line 258: „number of cells at the G1 phase appeared to increase with aging (Figure 4B).“).

3. I think it would be a highly valuable addition to this manuscript if the authors showcase the compatibility of their technique with CITE-seq approaches. This would help annotate rare cell types based on surface markers (a very classical approach in hematology) and increase throughput/multiplexing capacity via antibody-based hashing.

4. A recent preprint by Abay et al. (DOI: 10.1101/2024.03.27.587039) demonstrates the enrichment of rare-cell populations based on transcript expression vs. surface marker expression. It would be very valuable to discuss these developments,

including related work by some of the authors (e.g., FIND-seq) in the context of PURE-seq.

5. The authors should comment more on their cell recovery rate. For example, the recovery rate of ~30% in the species mixing experiment (584+112 cells recovered out of 2500 sorted cells) seems to be relatively low. In particular as minimizing cell loss by directly loading from the flow cytometer into the PIP-seq reaction for studying rare cells is one of the major advantages of the technique?

Minor comments

1. For the species mixing experiment it would be helpful to include an estimation for doublet rates for comparability to other methods as it appears to be high given the high mixing ratios and relatively low input cell concentrations.
2. The cell type annotation in Figure 3 could be further supported by additionally showing cell type-specific markers in the supplement (e.g., for pNeu, pMast, pER).

Reviewer #2

(Remarks to the Author)

Pan and colleagues creatively designed the PURE-seq to directly FACS sort rare cell populations for scRNA-seq using the recently published PIP-seq pipeline. The authors used the microfluidics-free system to minimize cell loss during the library preparation. PURE-seq can be a valuable tool after further validation. However, this manuscript has a few weak points. First, the authors did not perform side-by-side 10x library preparation post FACS sorting for the direct comparison. Second, the scRNA-seq of LT-HSCs is quite preliminary. The final conclusion is completely based on prediction without further validation and in-depth analysis.

Here are my major comments:

1. The author successfully enriched rare cells by mixing human-mouse cells and LT-HSC from mouse bone marrow. By comparing with the published scRNA-seq dataset, the authors demonstrated the capacity of PURE-seq to capture LT-HSC by sequencing fewer cells. Yet, given the complexity of 10x library preparation and technical differences between FACS sorter and operators, it is essential that the authors perform the same experimental procedures for 10x scRNA-seq and PIP-seq for comparison. Especially the major point of this manuscript is to compare PURE-seq with conventional 10x scRNA-seq post FACS sorting.
2. In Figure 3E, the authors identified a large population annotated as "unknown" after integrating the cell annotation according to the Herault et al dataset. Is this "unknown" population specific for middle-aged mice? What's the gene signature of this cluster? The authors need further analysis of this population.
3. In Figure S4, the authors re-clustered the LT-HSC and discovered 3 clusters. It is interesting that cluster 2, annotated as "rep" is absent in the old mice. What's the gene signature of this cluster? It seems cluster 2 in the middle-aged mice also express *Egr1* (Figure 5D). Is the absent of this cluster related to the aging process?
4. The manuscript ended with *Erg1* as a potential master regulator in hematopoiesis aging, yet the conclusion was based on the prediction of the DEG. It is a weak point of the manuscript without further in-depth analysis or validation. In the discussion, the authors vaguely mentioned HSC quiescent/senescence and age-related inflammation. Does *Erg1* upregulation regulate these pathways? Do you observe relevant gene signatures in your dataset?

Reviewer #3

(Remarks to the Author)

This study comprises a deep characterization of the transcriptomes in an enriched precursor cell fraction from murine bone marrow. Expression patterns were compared between young (8-12 wk), middle-aged (12 mo) and old (18 mo) mice. Cells (KLS_CD150+) were interrogated by single-cell RNAseq. PCA and UMAP visualization was compared and matched to an earlier baseline analysis of age-related transcriptomics in early cells (KLS) from Herault et al. The analysis here focused on *Egr1* expression, originally identified by Wagers et al. in 2008 in murine hemopoietic stem cells ("HSC") and linked with cell cycle and mobilization into blood. Similar association with cycling and aging was more recently reported in human normal and leukemic HSC and progenitors. *Egr1* was highlighted in the present study from among a list of ~20 genes whose transcript numbers in analytically defined hemopoietic stem cells ("HSC") varied with mouse age. The delivered items are single-cell RNAseq data on prior-enriched "HSC", identification of increased *Egr1* transcript numbers in samples from aged mice, and validation of the particular scRNAseq workflow used here including replacement of microfluidic droplet generation by emulsion encapsulation.

General comments on the strength of the approach and impact of the results:

Except for varying the ages of the sampled mice, the study is descriptive rather than experimental. Much of it essentially repeats similar explorations already published (e.g. Herault,5,7, numerous others). Proposed attribution of expression data to particular precursor stages rests on assumptions about heterogeneous, transcriptionally defined cells whose identity as "HSC" or long-term "LT-HSC" is not verified in functional assays. Informatically inferred functional relationships between transcription factors are not experimentally tested.

For a transcriptomic study to contribute trusted insight into regulation of aging in HSC, expression changes should be convincingly shown to occur in HSC themselves. Relevance of changes observed in more fleeting downstream progenitors would be uncertain. The required proof could be secured if RNAseq were performed on a purified fraction that had been

verified by biological assay to consist mainly of LT-HSC. No measurement of biological heterogeneity is documented in the present study - a major shortcoming - while the results indeed confirm considerable transcript profile heterogeneity in the fraction analyzed, including cell cycle position.

Although the analysis focused on cells clustered by PCA/UMAP with features consistent with LT-HSC identity, the uncertainty concerning functional heterogeneity of the input population leaves indeterminate which cells in the "LT-HSC" clusters were capable of permanent *in vivo* reconstitution and which were not. Further, the gating strategy used here (Kit⁺/Sca1⁺/CD150⁺/CD48⁻) is reported (e.g. 3) to yield only 10 - 20% of functionally defined LT-HSC among much higher numbers of shorter-term HSC. FACS labels that could have been incorporated to improve purity were not used in the present work (CD49b to segregate shorter term HSC, e.g. 4,6,8, and Rhodamine or CD34, e.g. 1,2,4, for sorting on cell cycle status). The expected (and demonstrated, e.g. Fig 3, especially 3E) heterogeneity of the purified fraction submitted to RNAseq makes attribution of changes difficult. For example, Figure 5E shows similar numbers of cells expressing low levels of Egr1 transcript at the 3 timepoints, and elevated numbers expressing high levels of Egr1. The result is interpreted as demonstrating elevated expression in aged "HSC", but the writing does not acknowledge the result is also compatible with lack of change in a low expressing "true" LT-HSC subcomponent. Increasing expression with age could as well be confined to more numerous advanced progeny. Changes in expression with age could reflect shifting proportions of LT-HSC among more advanced progeny. As conceived, the study is inconclusive on whether Egr1 expression does or does not change in functionally defined LT-HSC. The title and writing should properly be changed to reference the analyzed cells as "HSC and Progenitor, HSPC" cells rather than "LT-HSC".

Results of the study raise conceivable hypotheses concerning a potential role of Egr1 in HSPC components changing with time. The writing needs to better clarify how the new data on Egr1 adds to understanding of its role and site of expression in the precursor hierarchy, beyond raising unverified possibilities. More substantial support for the speculated role is needed from validating biological or other evidence. Despite the uncertainties, the utility of the data as they stand now might be better highlighted by a deeper and more informative discussion of the published evidence in human normal and leukemic cells and how the data here might fit or refute the roles already suggested for Egr1.

Specific comments

1. Single-cell RNAseq is performed here on FACS-sorted populations designed to isolate a protein-based phenotype used in earlier studies to enrich for LT-HSC. Although not acknowledged here, prior population enrichment, including of HSC, has become a standard for HSPC-focused 'omic studies (e.g. 5,6,7). The "PURE-seq" strategy in this study is not novel, the improved numbers of HSC available for analysis after purification are recognized in the field, and text implying novelty should be deemphasized.

2. The gating strategy in Fig S3A uses arbitrarily wide gates notably for Sca1 and CD150. The expected cluster of Kit^{hi}Sca1^{hi} cells reported in other studies is difficult to discern, but should be discrete and well populated after appropriate prior gating on other parameters (e.g. light scatter, lineage, CD34). Confining the sort more stringently to the Kit^{hi}and especially Sca1^{hi} cluster would reduce biological heterogeneity in the fraction submitted to RNAseq.

3. Despite isolation of cells on the basis of expression of cKit, Ly6a and CD150 proteins, it is troubling that all 3 transcripts were detected together in only 7% of sequenced cells. That may reflect deficient stringency in the sorting strategy, or more problematically insufficient capture of transcripts. Figure S3 shows only ~ 2000 genes detected per cell, and less than 4000 transcripts sequenced, suggesting that detection dropouts of expected genes must be frequent. Shallowness of capture is a known deficiency of present RNAseq workflows. The differing patterns in Figure 3B would be compatible with differing probabilities of transcript capture, greatest for c-Kit, lowest for Slamf1. Whatever the basis for the discrepancies between the sort parameters and detection of key markers in the RNAseq analysis, confidence in the precision of identity assignments is reduced in the current state of the RNAseq art. The issue deserves exposure and discussion in the paper - how immune are PCA assignments, or scGate clustering to this source of noise?

4. Figure 4 suggests considerable cycle phase heterogeneity in UMAP'd HSC. Evidence for involvement of Egr1 in cycle status has been reported before. It would be helpful to include a plot documenting the actual relationship between expression level of Egr1 and cycle state in the RNAseq data.

5. The pathway analysis performed here highlights ribosome synthesis and metabolic gene expression as the most prominent changes with age, as might be expected with changes in cell cycle distribution. A transcription factor network emanating from Egr1 (Fig 5C) derived from the present RNAseq data emerged as the "most probable" model from a transcription factor enrichment analysis. The description of the map needs elaboration. Exactly how much more probable was the illustrated model than competing models from the ChEA3 tool? Are the connections derived from known functional linkages? What details from the map suggested plausible biological roles of Egr1? Are any of these supported by published evidence?

6. In Figures 3E and 4F, "np1", "rep" and "tgf" need explanation in the legend.

References

1. CD34 expression by murine hematopoietic stem cells. Developmental changes and kinetic alterations. M Ogawa, Ann N Y Acad Sci. 938:139-45, 2001 PMID: 11458501
2. Dormant and Self-Renewing Hematopoietic Stem Cells and Their Niches. A Wilson, Ann. N.Y. Acad. Sci. 1106: 64–75, 2007 PMID: 17442778
3. Hematopoietic Stem Cells Reversibly Switch from Dormancy to Self-Renewal during Homeostasis and Repair. A Wilson, Cell 135:1118–1129, 2008 PMID: 19062086
4. Intermediate-term hematopoietic stem cells with extended but time-limited reconstitution potential. P Benveniste, Cell Stem Cell. 6:48–58, 2010 PMID: 20074534
5. Combined Single-Cell Functional and Gene Expression Analysis Resolves Heterogeneity within Stem Cell Populations. NK Wilson, Cell Stem Cell 16:712–724, 2015 PMID: 26004780
6. N-Cadherin-Expressing Bone and Marrow Stromal Progenitor Cells Maintain Reserve Hematopoietic Stem Cells. M Zhao, Cell Reports 26:652–669, 2019 PMID: 30650358
7. BloodSpot 3.0: a database of gene and protein expression data in normal and malignant haematopoiesis MH Gíslason, Nucleic Acids Research, 52(D1):D1138-D1142, 2024 PMID: 37933860
8. CD49b identifies functionally and epigenetically distinct subsets of lineage-biased hematopoietic stem cells. E Somuncular, Stem Cell Reports 17:1546-1560, 2022 PMID: 35714596

Version 1:

Reviewer comments:

Reviewer #1

(Remarks to the Author)

In their revised manuscript describing PURE-seq, the authors largely addressed the previous comments by this Reviewer. The additional data on circulating tumor cells is another solid showcase of the methodology. Still, the CNV analysis appears to be very noisy (healthy cells also appear to have multiple CNVs, which seems highly unlikely). This is likely attributable to low data sparsity. Some type of ground truth data of the primary melanoma would have been useful; otherwise, the low confidence nature of the CNV analysis should be transparently discussed.

Reviewer #2

(Remarks to the Author)

the authors have addressed my previous comments adequately. The manuscript describes a very useful new technique that can be applied to the study of small populations of cells in vivo and in vitro.

Reviewer #3

(Remarks to the Author)

This study characterizes and analyzes transcriptomes in an enriched precursor cell fraction from young, middle-aged and old mice. Age-related change in prevalence of Egr-1 transcripts is highlighted. A particular configuration of FACS sorting, cell capture and emulsion encapsulation, branded as 'PURE-seq', is also highlighted.

The submitted revision improves clarity of writing, softens a number of inadequately supported claims, and adds helpful context from the published literature. The manuscript however remains descriptive without functional testing of identities or proposed mechanisms. It is difficult to discern substantive advance over copious published work in this area, in technology, concept or understanding of precursor cell mechanisms.

REVIEWER COMMENTS

Reviewer #1 (Remarks to the Author):

In this manuscript, Pan et al., introduce and benchmark PURE-seq, a single-cell transcriptomics method with selective enrichment of cells prior to sequencing allowing for an increased capture of rare cells. The assay relies on the direct loading via FACS-sorting into PIP-seq reactions, thereby reducing cell loss. PURE-seq uses standard laboratory equipment which makes it more readily accessible. Finally, the authors apply PURE-seq to characterize LT-HSCs in murine aging and identify *Egr1* as a potential regulator in this context.

Overall, the manuscript is very well written, and the quality of the experiments and data is certainly very high.

We thank the reviewer for their thoughtful and positive appraisal of our manuscript.

This Reviewer has the following comments:

Major comments

1. Enriching rare cells and directly loading them into droplets to minimize loss is a very useful tool for the community and a helpful addition to the PIP-seq workflow. However, from a technical perspective, the combination of steps is quite logical, rather than presenting an innovative new protocol. In principle, a similar level of enrichment could be achieved with most other downstream single-cell genomic workflows. While the level of cell loss may be variable across platforms (e.g., including 10x), this has not been formally shown and is likely also only to be increasingly problematic with the increasing rarity of the cell population of interest.

As such, the authors could focus more on the technology providing hands-on guidelines such as how to fine-tune sorting settings for PURE-seq. For example, I assume optimal sorting settings are different for different cell types and the cell loading concentration could be quite variable depending on the prevalence of the cells of interest. Also, for rare cells more input material and long sorting times are necessary and therefore cells would have to remain viable in the FACS and downstream buffers. Therefore, I think it would be valuable to showcase the approach for another cell type/tissue such as, for example, circulating tumor cells (as the authors pointed out in the discussion), and discuss PURE-seq workflow parameters.

We thank the reviewer for this suggestion. To further demonstrate PURE-seq's utility, we have included data showcasing the application to circulating tumor cells (CTCs). This experiment involves n=2 peripheral blood samples of patients with metastatic melanoma and highlights PURE-seq's ability to capture transcriptomic data from CTCs efficiently (**Figure 2, Supplementary Figure 2**). These findings have been added to the manuscript.

To address the reviewer's suggestion of discussing PURE-seq workflow parameters, we have included an additional section "Optimization of Sorting Parameters" in the **Methods** outlining practical guidelines for fine-tuning sorting settings in PURE-seq. This includes recommendations on sorting precision modes, flow rates, and buffer compositions to optimize cell viability and capture efficiency for different cell types. Specifically, we emphasize that optimal sorting parameters, including flow rate and nozzle size, should be tailored based on cell size and fragility.

2. The biological novelty of the described findings appears rather limited and primarily serves the purpose of a showcase of the PURE-seq workflow. The observations regarding the increase in numbers of LT-HSCs, their gradual loss of self-renewal with aging as well as the identification of *EGR1* as key transcription factor do not seem very striking, and additional quantifications

should be shown. For example, the differences in percentage in flow and sequencing are rather subtle and are only shown for three data points with relatively low cell numbers and also without assessing variability between individual mice (e.g., Line 191: “percentage of LT-HSCs increased with age (Supplementary Figures 3A, B)”); Line 194: “(UMAP) plots that showed a higher number of hematopoietic cells [...] in the middle-aged and old samples compared to their young counterparts (Figure 3C)”); Line 258: „number of cells at the G1 phase appeared to increase with aging (Figure 4B).“).

We thank the reviewer for this thoughtful comment. We have now updated **Figure 3D** to include bar plots showing the percentages of LT-HSCs and non-LT-HSCs across age groups; of note, to avoid potential misclassification of these cells—given the absence of functional studies confirming their LT-HSC identity—we now refer to them based on their immunophenotypic markers (i.e., Sca1⁺Kit⁺Flt3⁻CD150⁺CD48⁻ HSCs).

Additionally, **Figures 4B and 4D** now present bar graphs depicting the total number of captured cells and Sca1⁺Kit⁺Flt3⁻CD150⁺CD48⁻ HSCs, respectively, across the different cell cycle phases. Regarding mouse-to-mouse variability, we acknowledge its importance in the context of hematopoietic cell biology. However, as outlined in the **Methods** section, our study used pooled samples for the following reasons:

1. Established inter-mouse variability in HSC frequencies has been well documented in previous studies comparing young and old mice. These include Dykstra et al.¹, Yamamoto et al.²; using donor mice from transplantation assays), Jang et al.³; using Pdzk1ip1-CreER; R26⁺Tom/Tom⁺ reporter mice), and Su et al.⁴ (incorporating the CD49b marker).
2. Pooling enabled precise cell numbers from a biologically variable bone marrow sample for downstream processing. Specifically, after lineage depletion, we directly sorted 2,500 Sca1⁺, Kit⁺, Flt3⁻, CD150⁺, CD48⁻ HSCs into PIP-seq tubes to immediately proceed with the PURE-seq protocol. This direct-sorting approach avoids post-sort pooling and preserves the integrity of the method.
3. The primary objective of our study is to demonstrate the capability of PURE-seq in enriching and sequencing rare cell populations as an alternative method to 10x single cell RNA sequencing (we have now included a comprehensive comparison to the 10x Genomics dataset from the manuscript by Héroult et al.⁵, which we used as a reference (please, see **Supplementary Figure 5**). To showcase the PURE-seq pipeline, we applied the method to both human (circulating melanoma cells from peripheral blood) and mouse (Sca1⁺, Kit⁺, Flt3⁻, CD150⁺, CD48⁻ HSCs from bone marrow) samples. **Figure Supplementary 3** provides a representative example of our Sca1⁺, Kit⁺, Flt3⁻, CD150⁺, CD48⁻ HSC sorting strategy (unmodified from our original manuscript submission). Each biological replicate was generated from pooled cells derived from 2–3 age-matched mice. In total, we included 2 replicates for the young group and 3 replicates each for the middle-aged and old groups.

3. I think it would be a highly valuable addition to this manuscript if the authors showcase the compatibility of their technique with CITE-seq approaches. This would help annotate rare cell types based on surface markers (a very classical approach in hematology) and increase

throughput/multiplexing capacity via antibody-based hashing.

We agree that incorporating antibody-based hashing and surface marker annotation would significantly expand the utility of PURE-seq. To address this, we cite evidence from the PIP-seq study, which has already demonstrated compatibility with CITE-seq to enable surface marker profiling in single-cell workflows. Since PURE-seq uses the same droplet emulsification principles and molecular compatibility as PIP-seq, these functionalities are readily transferrable. We have included this discussion in the revised manuscript and clarified that PURE-seq can directly integrate CITE-seq reagents to annotate rare cell populations while increasing throughput and multiplexing capacity. We have added the following language to the discussion:

“CITE-seq enables simultaneous measurement of surface protein markers and transcriptomes, which would help annotate rare cell types based on surface markers and increase throughput by multiplexing. Given that the PIP-seq workflow has previously demonstrated compatibility with CITE-seq¹², and PURE-seq is directly built upon the same framework, it is expected that PURE-seq will also be fully compatible with CITE-seq.”

4. A recent preprint by Abay et al. (DOI: 10.1101/2024.03.27.587039) demonstrates the enrichment of rare-cell populations based on transcript expression vs. surface marker expression. It would be very valuable to discuss these developments, including related work by some of the authors (e.g., FIND-seq) in the context of PURE-seq.

We thank the reviewer for highlighting the recent advancements in the enrichment of rare-cell populations based on transcript expression. These methods provide important perspectives on how PURE-seq could be further developed and adapted. To address this, we have expanded the discussion to acknowledge these complementary approaches and their potential integration with PURE-seq. Specifically, we have added the following section to the discussion:

“Recent advances in single-cell technologies have introduced innovative methods to enrich rare-cell populations based on transcriptomic features. PERFF-seq uses RNA FISH to target specific transcripts, enabling the profiling of fixed nuclei, including those from FFPE samples. While PURE-seq, like PIP-seq, is compatible with DSP-methanol fixation, it has not yet been adapted for FFPE samples. Incorporating RNA FISH-based enrichment from PERFF-seq into PURE-seq could expand its applicability to challenging fixed tissues. Similarly, FIND-seq’s nucleic acid cytometry using microfluidic droplets offers another potential avenue for adaptation. By integrating FIND-seq’s enrichment principle into PURE-seq’s microfluidics-free workflow, it may be possible to achieve nucleic acid-based enrichment without the need for complex platforms. Together, these approaches demonstrate how PURE-seq could evolve to address a broader range of sample types and rare-cell applications.”

5. The authors should comment more on their cell recovery rate. For example, the recovery rate of ~30% in the species mixing experiment (584+112 cells recovered out of 2500 sorted cells) seems to be relatively low. In particular as minimizing cell loss by directly loading from the flow cytometer into the PIP-seq reaction for studying rare cells is one of the major advantages of the technique?

We thank the reviewer for highlighting the importance of discussing cell recovery rates in the context of PURE-seq. The ~30% recovery rate in the species mixing experiment reflects the combined efficiency of two steps: FACS sorting and the PIP-seq capture chemistry. This experiment was performed using an earlier version of the PIP-seq chemistry, which had a capture efficiency of ~40% (Fluent Biosciences, FB0001026, 20210908 version). When

combined with a typical FACS recovery rate at ~80%, this resulted in an overall recovery rate of ~30%.

It is worth noting that the latest version of PIP-seq chemistry has improved capture efficiency to ~60%⁶, which, when integrated with PURE-seq, could enhance the overall recovery rate to ~50%. Additionally, FACS recovery rates could also be further optimized by adjusting sorting parameters and collection conditions. These advancements are expected to significantly reduce cell loss and further strengthen PURE-seq's utility for studying rare-cell populations. We have clarified these points in the revised text to provide a more comprehensive discussion of recovery rates and the potential for future improvements.

Minor comments

1. For the species mixing experiment it would be helpful to include an estimation for doublet rates for comparability to other methods as it appears to be high given the high mixing ratios and relatively low input cell concentrations.

We thank the reviewer for pointing out the importance of comparing our doublet rates to other methods. In the table below, the mixing rate is the observed fraction of the mixed cells in our species mixing experiments and the inferred multiplet rate is twice the mixing rate (Macosko et al.⁷; Zheng et al.⁸).

Experiment	# of Cells Recovered	# of Mixed Cells	Mixing Rate (%)	Multiplet Rate (%)
1	703	7	1.00	1.99
2	560	7	1.25	2.50
3	318	2	0.63	1.26
4	131	0	0.00	0.00
5	65	5	7.69	15.38
6	8	0	0.00	0.00

Across our 6 experiments, the multiplet rate ranged from 0% to 15%, with 5 experiments demonstrating multiplet rates below 3%. Since PURE-seq was based on PIP-seq, PIP-seq originally reported a multiplet rate of 6% (Clark et al.⁹). The relatively higher multiplet rate observed in one of our experiments (15%) was most likely due to slow pre-lysis of cells by proteinase K at low temperature during FACS sorting. Notably, more recent versions of PIP-seq have addressed this by removing proteinase K lysis reagents from reaction tubes. Instead, chemical lysis reagents are now added after cell-loading, significantly reducing the multiplet rate to below 2%.

While this adjusted rate is still somewhat higher than typical multiplet rates reported for 10X Genomics platforms (~0.8% at ~1,000 cells and ~0.4% at ~500 cells recovered), PIP-seq and PURE-seq offer superior scalability. Specifically, we demonstrated that our method can effectively recover fewer than 10 cells per experiment, a scale that remains challenging for 10X Genomics workflows.

2. The cell type annotation in Figure 3 could be further supported by additionally showing cell type-specific markers in the supplement (e.g., for pNeu, pMast, pER).

We thank the reviewer for this comment. In response, we have now included additional markers such as *Elane* (pMast, pNeu), *Klf1* (pER), and *Hdc* (pMast, pNeu) (see **Supplementary Figure 5A**). These markers were also reported in the study by Heurault et al.⁵, which we used as our 10x Genomics reference.

Reviewer #2 (Remarks to the Author):

Pan and colleagues creatively designed the PURE-seq to directly FACS sort rare cell populations for scRNA-seq using the recently published PIP-seq pipeline. The authors used the microfluidics-free system to minimize cell loss during the library preparation. PURE-seq can be a valuable tool after further validation. However, this manuscript has a few weak points. First, the authors did not perform side-by-side 10x library preparation post FACS sorting for the direct comparison. Second, the scRNA-seq of LT-HSCs is quite preliminary. The final conclusion is completely based on prediction without further validation and in-depth analysis.

Here are my major comments:

1. The author successfully enriched rare cells by mixing human-mouse cells and LT-HSC from mouse bone marrow. By comparing with the published scRNA-seq dataset, the authors demonstrated the capacity of PURE-seq to capture LT-HSC by sequencing fewer cells. Yet, given the complexity of 10x library preparation and technical differences between FACS sorter and operators, it is essential that the authors perform the same experimental procedures for 10x scRNA-seq and PIP-seq for comparison. Especially the major point of this manuscript is to compare PURE-seq with conventional 10x scRNA-seq post FACS sorting.

We appreciate the reviewer's thoughtful comments. As the reviewer noted, our primary objective in this manuscript was not to directly benchmark PURE-seq against 10X Genomics methods. Rather, our goal was to develop PURE-seq as a novel approach specifically designed to address technical gaps in studying rare cell populations, particularly populations whose abundance is below the minimal cell input requirements of current droplet-based microfluidics approaches. For example, PURE-seq was explicitly developed to process populations at extremely low abundance (i.e. rarity of 1 in 10,000, 1 in 100,000, 1 in 1,000,000) or low input (i.e. <100 cells), scenarios in which conventional droplet-based technologies like 10X Genomics are technically impractical or infeasible.

We acknowledge the importance of a side-by-side comparison with 10X Genomics data. However, generating new 10X data would introduce potential confounding factors, such as batch effects resulting from differences in mouse cohorts and sequencing runs. We consider these sources of variability comparable to those present in publicly available datasets. Therefore, to ensure a rigorous and controlled comparison, we utilized the 10X scRNA-seq HSPC dataset from Héroult et al.⁵, with deeper analysis of LT-HSCs, as the 10X Genomics reference for our study. Yet, as a reminder, we have refrained from using the term 'LT-HSCs' in the newly submitted version of our manuscript and instead refer to these cells solely by their immunophenotypic markers: *Sca1*⁺, *Kit*⁺, *Flt3*⁻, *CD150*⁺, *CD48*⁻ HSCs.

Importantly, we replicated the lineage depletion and cell-sorting pipeline described in that study prior to sequencing our own samples. We then integrated our PURE-seq dataset with the published 10X dataset to enable a robust analytical comparison (**Supplementary Figure 4**). We observed similar marker gene expression patterns between the datasets and high correlation of the genes by cluster. These considerations, along with the results of the integrative analysis,

have been clearly addressed in the revised manuscript to clarify our rationale, highlight the technical advantages of PURE-seq, and contextualize our findings alongside the 10X dataset.

2. In Figure 3E, the authors identified a large population annotated as “unknown” after integrating the cell annotation according to the Herault et al dataset. Is this “unknown” population specific for middle-aged mice? What’s the gene signature of this cluster? The authors need further analysis of this population.

We thank the reviewer for raising this point. Upon careful re-examination, we realized that the confusion arose from an unclear annotation in the initial version of Figure 3E. In our updated analysis which involved integration of the Herault et al. dataset, we did not detect a distinct “unknown” cluster unique to the middle-aged mice. The “unknown” labeling originally indicated cells that could not be confidently annotated based on initial clustering, rather than representing a specific biological population unique to middle-aged mice.

In the revised manuscript, we clarified this point by presenting an updated integrated clustering result, which demonstrates clear, consistent annotation across all clusters, eliminating the presence of any ambiguous or “unknown” populations. This ensures accurate biological interpretation and addresses the reviewer’s concern directly.

3. In Figure S4, the authors re-clustered the LT-HSC and discovered 3 clusters. It is interesting that cluster 2, annotated as “rep” is absent in the old mice. What’s the gene signature of this cluster? It seems cluster 2 in the middle-aged mice also express *Egr1* (Figure 5D). Is the absent of this cluster related to the aging process?

We thank the reviewer for this insightful question. However, in the revised manuscript, we addressed Reviewer 3, Comment 3 by implementing scGate¹⁰ as our method for *Sca1*⁺, *Kit*⁺, *Flt3*⁻, *CD150*⁺, *CD48*⁻ HSC selection, rather than using it as an orthogonal validation as we did in our original manuscript version. With scGate, we applied a gating strategy based on established markers (*Sca1*⁺, *Kit*⁺, *Flt3*⁻, *CD150*⁺, and *CD48*⁻). This approach enabled us to identify approximately 52% of the cells as *Sca1*⁺*Kit*⁺*Flt3*⁻*CD150*⁺*CD48*⁻ HSCs, offering a more accurate recovery of the expected population compared to our original thresholding method, which was more stringent and prone to stochastic transcript dropout.

After subsetting and reclustering the scGate-identified *Sca1*⁺*Kit*⁺*Flt3*⁻*CD150*⁺*CD48*⁻ HSCs, we observed 10 distinct clusters, in contrast to the 3 clusters initially reported. Although we noted differences in age ratios and cell numbers across clusters (**Supplementary Figure 4**), we did not observe the loss of any clusters with aging. We have removed statements referring to a loss of heterogeneity in the revised manuscript.

4. The manuscript ended with *Erg1* as a potential master regulator in hematopoiesis aging, yet the conclusion was based on the prediction of the DEG. It is a weak point of the manuscript without further in-depth analysis or validation. In the discussion, the authors vaguely mentioned HSC quiescent/senescence and age-related inflammation. Does *Erg1* upregulation regulate these pathways? Do you observe relevant gene signatures in your dataset?

We sincerely thank the reviewer for raising this thoughtful concern. We acknowledge that the title and emphasis on *Egr1* in parts of the manuscript may have unintentionally conveyed the impression that elucidating *Egr1*’s mechanistic role was the central aim of our study. We regret any confusion this may have caused. Our primary objective is to introduce and validate the PURE-seq pipeline as a versatile and efficient method for enriching rare cell populations from

both human (peripheral blood) and mouse (bone marrow) tissues, enabling their direct integration into the PIP-seq workflow. PURE-seq yields results comparable to the 10X Genomics platform, while requiring fewer cells and reducing sample handling. In light of this technical focus, we believe that a detailed mechanistic investigation of *Egr1* falls outside the scope of the present work.

To better clarify the intent of our study, we have revised the manuscript accordingly—modifying the title, abstract, and relevant sections of the introduction and discussion—and further strengthened our technical demonstration by including an additional dataset from human peripheral blood (**Figure 2**). Regarding the observed upregulation of *Egr1*, we agree that this is a compelling observation and have expanded our analysis to underscore its potential significance. As presented in **Supplementary Figure 6**, we performed unbiased gene enrichment analyses using ShinyGO, comparing *Egr1*^{low} and *Egr1*^{high} Sca1⁺Kit⁺Flt3⁻CD150⁺CD48⁻ HSCs across different age groups (**panel F**). Across all age groups, *Egr1*^{high} cells consistently show enrichment for inflammatory pathways, including TNF α signaling via NF- κ B. Notably, this pathway is also significantly enriched in the comparison between bulk old and young Sca1⁺Kit⁺Flt3⁻CD150⁺CD48⁻ HSCs (**panel G**), independent of *Egr1* expression levels.

While we explored the possibility of senescence-related pathway involvement, we did not observe robust enrichment and therefore chose not to include it in the manuscript (**Figure 1 of this rebuttal letter**). We believe that a deeper exploration of senescence and *Egr1* would require investigation of interactions between *Egr1*^{high} Sca1⁺Kit⁺Flt3⁻CD150⁺CD48⁻ HSCs and other bone marrow components—such as stromal¹¹, progenitor¹², or differentiated¹³ cells—as they have been shown to have a senescent profile during aging in previous human or mouse studies. Altogether, these findings suggest that *Egr1* may contribute to HSC aging, though we respectfully propose that in-depth mechanistic validation be pursued in future studies, given the technical scope of our current manuscript.

Figure 1 | Gene set enrichment analysis (GSEA) of senescence-associated signatures in *Egr1*^{high} versus *Egr1*^{low} Sca1⁺Kit⁺Flt3⁻CD150⁺CD48⁻ HSCs. GSEA plots showing enrichment of

the Jackson laboratory senescence gene signature (left) and the Reactome senescence-associated secretory phenotype (SASP) pathway (right).

Reviewer #3 (Remarks to the Author):

This study comprises a deep characterization of the transcriptomes in an enriched precursor cell fraction from murine bone marrow. Expression patterns were compared between young (8-12 wk), middle-aged (12 mo) and old (18 mo) mice. Cells (KLS_CD150+) were interrogated by single-cell RNAseq. PCA and UMAP visualization was compared and matched to an earlier baseline analysis of age-related transcriptomics in early cells (KLS) from Haurat et al. The analysis here focused on Egr1 expression, originally identified by Wagers et al. in 2008 in murine hemopoietic stem cells ("HSC") and linked with cell cycle and mobilization into blood. Similar association with cycling and aging was more recently reported in human normal and leukemic HSC and progenitors. Egr1 was highlighted in the present study from among a list of ~20 genes whose transcript numbers in analytically defined hemopoietic stem cells ("HSC") varied with mouse age. The delivered items are single-cell RNAseq data on prior-enriched "HSC", identification of increased Egr1 transcript numbers in samples from aged mice, and validation of the particular scRNAseq workflow used here including replacement of microfluidic droplet generation by emulsion encapsulation.

General comments on the strength of the approach and impact of the results:

Except for varying the ages of the sampled mice, the study is descriptive rather than experimental. Much of it essentially repeats similar explorations already published (e.g. Haurat,5,7, numerous others). Proposed attribution of expression data to particular precursor stages rests on assumptions about heterogeneous, transcriptionally defined cells whose identity as "HSC" or long-term "LT-HSC" is not verified in functional assays. Informatically inferred functional relationships between transcription factors are not experimentally tested.

For a transcriptomic study to contribute trusted insight into regulation of aging in HSC, expression changes should be convincingly shown to occur in HSC themselves. Relevance of changes observed in more fleeting downstream progenitors would be uncertain. The required proof could be secured if RNAseq were performed on a purified fraction that had been verified by biological assay to consist mainly of LT-HSC. No measurement of biological heterogeneity is documented in the present study - a major shortcoming - while the results indeed confirm considerable transcript profile heterogeneity in the fraction analyzed, including cell cycle position.

Although the analysis focused on cells clustered by PCA/UMAP with features consistent with LT-HSC identity, the uncertainty concerning functional heterogeneity of the input population leaves indeterminate which cells in the "LT-HSC" clusters were capable of permanent in vivo reconstitution and which were not. Further, the gating strategy used here (Kit+/Sca1+/CD150+/CD48-) is reported (e.g. 3) to yield only 10 - 20% of functionally defined LT-HSC among much higher numbers of shorter-term HSC. FACS labels that could have been incorporated to improve purity were not used in the present work (CD49b to segregate shorter term HSC, e.g. 4,6,8, and Rhodamine or CD34, e.g. 1,2,4, for sorting on cell cycle status). The expected (and demonstrated, e.g. Fig 3, especially 3E) heterogeneity of the purified fraction submitted to RNAseq makes attribution of changes difficult. For example, Figure 5E shows similar numbers of cells expressing low levels of Egr1 transcript at the 3 timepoints, and elevated numbers expressing high levels of Egr1. The result is interpreted as demonstrating

elevated expression in aged "HSC", but the writing does not acknowledge the result is also compatible with lack of change in a low expressing "true" LT-HSC subcomponent. Increasing expression with age could as well be confined to more numerous advanced progeny. Changes in expression with age could reflect shifting proportions of LT-HSC among more advanced progeny. As conceived, the study is inconclusive on whether *Egr1* expression does or does not change in functionally defined LT-HSC. The title and writing should properly be changed to reference the analyzed cells as "HSC and Progenitor, HSPC" cells rather than "LT-HSC".

Results of the study raise conceivable hypotheses concerning a potential role of *Egr1* in HSPC components changing with time. The writing needs to better clarify how the new data on *Egr1* adds to understanding of its role and site of expression in the precursor hierarchy, beyond raising unverified possibilities. More substantial support for the speculated role is needed from validating biological or other evidence. Despite the uncertainties, the utility of the data as they stand now might be better highlighted by a deeper and more informative discussion of the published evidence in human normal and leukemic cells and how the data here might fit or refute the roles already suggested for *Egr1*.

We thank the reviewer for their thoughtful and detailed critique of our manuscript. We fully acknowledge the limitations raised, particularly concerning the functional heterogeneity of the sorted HSPC population and the need for caution when attributing transcriptional changes—such as *Egr1* upregulation—specifically to bona fide long-term HSCs. We agree that, in the absence of functional validation, definitive classification of LT-HSCs based solely on surface markers is inherently limited. In response to this important point, we have revised the manuscript to avoid overinterpretation of our data. Specifically, we have changed the title to "PURE-seq: Integrating FACS and Particle-templated Instant Partitioning Sequencing for Single-Cell Genomics of Rare Cells" to better reflect the technical focus of our study. Additionally, throughout the manuscript, we now refer to the analyzed population as "Sca1⁺Kit⁺Flt3⁻CD150⁺CD48⁻ HSCs" rather than "LT-HSCs", and have removed or rephrased any statements implying functional identity or long-term repopulating potential.

Our primary goal in this study is to introduce and validate PURE-seq as a flexible, accessible single-cell transcriptomics pipeline for profiling rare cell populations, rather than to define a mechanistic role for *Egr1* in HSC aging. That said, we agree that *Egr1* emerged as an intriguing candidate, and our data suggest its expression increases with age within the Sca1⁺Kit⁺Flt3⁻CD150⁺CD48⁻ HSC compartment. We fully acknowledge that our study does not resolve whether this change occurs within true LT-HSCs or reflects a shift in subpopulation proportions. To reflect this, we have explicitly discussed the alternative interpretation—insightfully raised by the reviewer—that the observed increase in *Egr1*^{high} cells may be due to an increased representation of *Egr1*-expressing progenitors, rather than an intrinsic upregulation within LT-HSCs. However, upon further examination of *Egr1* expression in relation to cell cycle status within our dataset, we observed a negative correlation between *Egr1* expression and cell proportions in G2/M and S phases (**Figure 5G**). This finding supports the interpretation that *Egr1*^{high} cells are less proliferative.

In addition, we have expanded the discussion to place our findings within the context of prior work on *Egr1* in both normal and leukemic hematopoiesis (**please see below in cursive**). While we intentionally refrain from drawing strong mechanistic conclusions, we hope this contextualization demonstrates how our PURE-seq-derived dataset can help generate testable hypotheses for future functional studies, which aligns with the intended scope of this manuscript.

“Using PURE-seq in combination with the ChEA3-derived prediction (Figure 5 C and D), we identified Egr1 as a putative master regulator of HSC aging in mice—specifically within the Sca1⁺Kit⁺Flt3⁻CD150⁺CD48⁻ HSC population—supporting previous evidence in the field and providing a foundation for future genomic, epigenomic, and mechanistic studies. The role of Egr1 as a TF relevant to murine HSC function was first described by Min et al. (2008), who identified Egr1 as essential for maintaining HSC quiescence and retention within the stem cell niche. More recently, Stoddart et al. (2022) demonstrated that Egr1 regulates genes involved in differentiation, inflammation, and DNA damage responses. They also showed that Egr1 haploinsufficiency provides a competitive advantage to HSPCs subjected to stress from serial bone marrow transplantation, consistent with its previously described role as a haploinsufficient myeloid leukemia gene in the context of N-ethyl-nitrosourea-induced mutagenesis. Interestingly, Egr1 expression increases in middle age in our dataset (Figure 5 A and F), suggesting a potential role as an early master regulator of HSC aging. This finding supports the idea that hematopoietic aging begins in middle age³⁹ and may point towards a functional link between age-associated hematopoietic DEG and myeloid cell transformation. It also underscores the need for future studies investigating how the temporal dynamics of gene expression contribute to HSC aging. Yet, it remains essential to functionally characterize HSPC subpopulations in relation to Egr1 expression, as observed changes may reflect shifts in the proportion of HSCs differentiating into progenitors or a more lineage-restricted progeny expressing Egr1. Nonetheless, we hypothesize that these changes probably originate within the HSC compartment, based on the observed negative correlation between Egr1 expression and the proportion of cells in the G2/M and S phases of the cell cycle (Figure 5G). This relationship suggests that Egr1^{high} cells are less proliferative, a characteristic more consistent with bona fide HSCs. However, this interpretation remains speculative and warrants functional validation to determine the precise role of Egr1 in HSC aging”.

Regarding the FACS gating strategy, we agree with the reviewer that additional markers (e.g., CD49b, CD34, Rhodamine) could have enhanced the purity of LT-HSCs. However, our aim was to prioritize simplicity and accessibility to enable broader adoption of the PURE-seq pipeline, including by researchers studying non-hematopoietic tissues or non-HSPC subsets. We therefore selected a widely used gating strategy¹⁴ that has also been applied in recent studies, such as that by Kucinski et al.¹⁵, for profiling HSPC subpopulations in single-cell analyses. In response to the reviewer’s comment, we have revised the manuscript to explicitly acknowledge the limitations of our approach and to encourage users to refine marker panels according to their specific biological questions.

In summary, we believe our study provides a robust technical foundation for future investigations into rare cell populations using PURE-seq, while recognizing that further biological validation—particularly in relation to *Egr1* function and HSC heterogeneity—is an important next step in HSC aging-focused studies.

Specific comments

1. Single-cell RNAseq is performed here on FACS-sorted populations designed to isolate a protein-based phenotype used in earlier studies to enrich for LT-HSC. Although not acknowledged here, prior population enrichment, including of HSC, has become a standard for

HSPC-focused 'omic studies (e.g. 5,6,7). The "PURE-seq" strategy in this study is not novel, the improved numbers of HSC available for analysis after purification are recognized in the field, and text implying novelty should be deemphasized.

We thank the reviewer for this constructive comment, which adds important context and rigor to our study by highlighting relevant prior literature. In response, we have carefully revised the manuscript to avoid language implying undue novelty and have explicitly clarified that the primary contribution of our work lies in the integration of FACS and PIP-seq (PURE-seq) to generate high-quality single-cell transcriptomic datasets from rare cell populations. This technical advance, rather than a novel biological discovery, is the central focus and innovation of our study.

2. The gating strategy in Fig S3A uses arbitrarily wide gates notably for Sca1 and CD150. The expected cluster of Kit^{hi}Sca1^{hi} cells reported in other studies is difficult to discern, but should be discrete and well populated after appropriate prior gating on other parameters (e.g. light scatter, lineage, CD34). Confining the sort more stringently to the Kit^{hi} and especially Sca1^{hi} cluster would reduce biological heterogeneity in the fraction submitted to RNAseq.

As stated earlier, we agree with the reviewer that our gating strategy could have been further refined by incorporating additional markers to increase the purity of the sorted population. However, our intention was to prioritize simplicity and accessibility, particularly to demonstrate that PURE-seq can yield high-quality single-cell transcriptomic data even with a minimal and commonly used gating panel. For this reason, we adopted a straightforward strategy—also employed in recent studies, including Kucinski et al.¹⁵—to better showcase the robustness and versatility of the PURE-seq pipeline. Importantly, we compared our data to a similarly gated 10X Genomics dataset (Hérault et al.⁵), which allowed for a meaningful side-by-side evaluation. That said, we fully acknowledge the reviewer's point and have revised the manuscript to transparently discuss this limitation and to note that additional markers (e.g., CD34, CD49b, Rhodamine) could be used in future applications to improve population purity (**please see below in cursive**).

“Although we employed here a simplified gating strategy based on expression of SLAM family markers (CD150⁺CD48⁻), which has also been recently used by others for sorting HSPC populations for single-cell genomics³³, it is important to note that additional markers can be incorporated to refine the isolation of functional HSCs. For instance, CD49b expression distinguishes myeloid-biased (CD49b⁻) from lymphoid-biased (CD49b⁺) HSCs and has recently been shown to remain conserved with aging³⁴. Further, the inclusion of CD34 and CD135 (encoded by Flt3) as negative selection markers can further enrich for HSCs, which are characterized by a CD34⁻CD135⁻ phenotype³⁵.”

We also acknowledge the reviewer's insightful suggestion regarding the importance of defining a more stringent Kit^{high}, Sca1^{high} population, which we already aimed to address, as shown in **Supplementary Figure 3, Panel A**. **Supplementary Figure 3A and 3B** present representative FACS plots from our pooled bone marrow samples and have not been modified from our originally submitted version. However, we have removed interpretive commentary associated with these plots in the main text and now refer to them simply as representative examples of the gating strategy used, instead of making any biological observation from them, which would not be scientifically rigorous.

3. Despite isolation of cells on the basis of expression of cKit, Ly6a and CD150 proteins, it is

troubling that all 3 transcripts were detected together in only 7% of sequenced cells. That may reflect deficient stringency in the sorting strategy, or more problematically insufficient capture of transcripts. Figure S3 shows only ~ 2000 genes detected per cell, and less than 4000 transcripts sequenced, suggesting that detection dropouts of expected genes must be frequent. Shallowness of capture is a known deficiency of present RNAseq workflows. The differing patterns in Figure 3B would be compatible with differing probabilities of transcript capture, greatest for c-Kit, lowest for Slamf1. Whatever the basis for the discrepancies between the sort parameters and detection of key markers in the RNAseq analysis, confidence in the precision of identity assignments is reduced in the current state of the RNAseq art. The issue deserves exposure and discussion in the paper - how immune are PCA assignments, or scGate clustering to this source of noise?

We appreciate the reviewer's insightful comment regarding the limited detection of Kit, Ly6a, and Slamf1 transcripts (~7% co-detection) despite protein-level sorting. This phenomenon is well recognized in single-cell RNA sequencing, where dropout is an inherent limitation due to the stochastic capture of transcripts. In our initial analysis, we relied on direct thresholding of individual gene expression, a strategy that is highly susceptible to dropout and likely underestimated the frequency of marker-positive cells. In the original version of the manuscript, we only applied scGate as an alternative method to visualize purity of Sca1⁺Kit⁺Fit3⁻CD150⁺CD48⁻ HSC.

To address this limitation, we have now adopted scGate as our primary method for Sca1⁺Kit⁺Fit3⁻CD150⁺CD48⁻ HSC identity assignment throughout the revised analysis. scGate is a method specifically designed to address the sparsity of single-cell data by combining rank-based UCell scoring with local k-nearest neighbor smoothing. This enables the identification of marker-positive cells based on aggregate gene signature activity, rather than requiring simultaneous detection of all marker transcripts. While our dataset did not include a companion antibody-derived tag (ADT) panel, the CITE-seq ADT data presented in the original scGate publication demonstrated that the method maintains high precision and recall despite scRNA-seq data sparsity¹⁰. For identifying LT-HSCs, we used a gating model based on well-established markers (Sca1⁺, Kit⁺, Fit3⁻, CD150⁺, and CD48⁻), as defined in previous studies¹⁶. Using this approach, we were able to identify ~52% of cells as LT-HSCs, demonstrating a more faithful recovery of the expected cell population than with the original thresholding method.

Regarding data quality, we would like to clarify that our dataset is in line with expected performance for droplet-based scRNA-seq platforms. To directly benchmark our data quality, we performed an integrated analysis with the Herculot et al. 10X dataset on HSPCs. This comparison showed that our dataset exhibits comparable quality metrics, including the number of detected genes per cell (nFeature_RNA), total transcript counts (nCount_RNA), as well as mitochondrial and ribosomal content [**Supplementary Figure 3F (dataset before subsetting) and Supplementary Figure 4C (dataset after subsetting and reclustering Sca1⁺Kit⁺Fit3⁻CD150⁺CD48⁻ HSC HSCs)**]. These findings indicate that the observed dropouts reflect the general sensitivity limitations of current scRNA-seq platforms rather than any inadequacy of our capture method.

We have revised the manuscript to explicitly include these points, highlighting how our refined scGate approach and benchmarking against external datasets support the reliability of our cell-type assignments.

4. Figure 4 suggests considerable cycle phase heterogeneity in UMAP'd HSC. Evidence for involvement of Egr1 in cycle status has been reported before. It would be helpful to include a

plot documenting the actual relationship between expression level of *Egr1* and cycle state in the RNAseq data.

We thank the reviewer for this insightful comment. To address the reviewer's previous comment, we adopted the scGate algorithm to more precisely select *Sca1*⁺*Kit*⁺*Flt3*⁻*CD150*⁺*CD48*⁻ HSC HSCs for downstream analysis. This refinement improved the resolution of cell cycle phase assignment, as now reflected in the updated **Figures 4B and 4D**.

To directly address the reviewer's request, we analyzed the relationship between *Egr1* expression levels and cell cycle state within this refined HSC subset. Specifically, we stratified *Egr1* expression into three bins—low (0–1), mid (1–1.5), and high (>1.5)—and quantified the distribution of cells in G1, S, and G2/M phases across these expression levels. This analysis revealed a negative correlation between high *Egr1* expression and G2/M and S phases, suggesting that elevated *Egr1* levels might be associated with reduced proliferative activity. Interestingly, this observation might be consistent with findings from a human study¹⁷, which reported that *EGR1* is upregulated in aging HSCs independently of cell cycle phase yet also showed altered cell cycle dynamics specifically during S phase. Our results in murine *Sca1*⁺*Kit*⁺*Flt3*⁻*CD150*⁺*CD48*⁻ HSCs might reflect a similar phenomenon and further support the notion that *Egr1* expression and cell cycle regulation are linked during aging. Furthermore, our observations are consistent with the seminal findings of Min et al.¹⁸, who demonstrated in murine hematopoiesis that reduced *Egr1* expression is associated with a more proliferative state of LT-HSCs and a more recent manuscript that shows that a downregulation of *EGR1* promotes HSC self-renewal¹⁹.

4. The pathway analysis performed here highlights ribosome synthesis and metabolic gene expression as the most prominent changes with age, as might be expected with changes in cell cycle distribution. A transcription factor network emanating from *Egr1* (Fig 5C) derived from the present RNAseq data emerged as the "most probable" model from a transcription factor enrichment analysis. The description of the map needs elaboration. Exactly how much more probable was the illustrated model than competing models from the ChEA3 tool? Are the connections derived from known functional linkages? What details from the map suggested plausible biological roles of *Egr1*? Are any of these supported by published evidence?

We appreciate the reviewer's comment. We have now expanded the ChEA3 section in the revised manuscript (**please see below in cursive**). ChEA3 is a web-based transcription factor (TF) enrichment analysis tool designed to prioritize candidate TFs that are most likely to regulate the observed gene expression changes. It integrates multiple data sources, including ChIP-seq datasets from ENCODE, ReMap, and individual studies; co-expression analyses derived from processed RNA-seq data (e.g., GTEx and ARCHS4); and curated libraries such as Enrichr and the ENCODE TF resource.

*"Next, using a web-based transcription factor (TF) enrichment analysis tool, ChEA3⁴⁴, which predicts the most probable TF network underlying gene expression changes based on existing epigenetic and transcriptomic datasets, we identified *EGR1* as a core node of the resulting TF network driving the transcriptional upregulation in old *Sca1*⁺*Kit*⁺*Flt3*⁻*CD150*⁺*CD48*⁻ HSCs (Figure 5C), as *EGR1* emerged as the top-ranked hit (Figure 5D). As an orthogonal validation,*

we performed ChEA3 analysis using a DEG list previously generated by Young et al. (2022) from sorted *Lin⁻Sca1⁺Kit⁺EPCR⁺Flt3⁻CD150⁺CD48⁻CD41⁺* HSCs—representing LT-HSCs in their manuscript—from young and old mice. *EGR1* again appeared as a central component of the predicted TF network (**Supplementary Figure 6A**), ranking within the top 20 hits (**Supplementary Figure 6B**). In parallel, we conducted ChEA3 analyses using upregulated genes from the old vs. young and old vs. middle-aged comparisons. *EGR1* was identified in the resulting TF networks from both analyses (**Supplementary Figure 6C**). These findings suggest that *Egr1* is involved in transcriptional changes across the entire aging spectrum, including within advanced age (i.e., from middle-aged to old), in murine HSCs.”

In our analysis, we employed the mean rank aggregation method across all available databases, as it has been shown through benchmarking to yield the most robust and reliable TF predictions²⁰. This approach generated a ranked list of TFs predicted to be key regulators of the transcriptional changes in our dataset. From this list, ChEA3 also constructed a regulatory network, highlighting interactions among the top-ranked TFs. Notably, *Egr1* emerged as the highest-ranking candidate and appears to form a regulatory module with other highly ranked TFs, including *Fosb* and *Jun* (**Figure 2 of this rebuttal letter; it coincides with Figure 5, panel D of our newly submitted version**).

We clarify that ChEA3 does not provide competing models or causal predictions; rather, it outputs a prioritized list of TFs based on statistical enrichment and integration of multiple datasets. Regarding the functional links between *Egr1* and other TFs identified in our analysis, Stoddart et al.²¹ provide supporting evidence. Using STRING pathway analysis, they demonstrated interactions between *EGR1*, *FOS*, and *JUN*, and further showed that *EGR1* can stimulate the transcription of *FOS/JUN* network genes in functional assays.

Additionally, their study reported direct binding of *EGR1* to the promoters of *HEY1/2*, among other genes implicated in hematopoietic self-renewal and differentiation using human HSPCs. Notably, *Hey2* also appears among the transcription factors ranked in our ChEA3 analysis, further supporting a potential regulatory module centered around *Egr1*.

Figure 2 | ChEA3-predicted transcription factors potentially driving upregulated gene expression in aged Sca1⁺Kit⁺Flt3⁻CD150⁺CD48⁻ HSCs compared to young and middle-aged counterparts.

6. In Figures 3E and 4F, "np1", "rep" and "tgf" need explanation in the legend.

We thank the reviewer for this comment. We have revised the legends and explained all the applicable abbreviations accordingly.

References

1. CD34 expression by murine hematopoietic stem cells. Developmental changes and kinetic alterations.
M Ogawa, Ann N Y Acad Sci. 938:139-45, 2001 PMID: 11458501
2. Dormant and Self-Renewing Hematopoietic Stem Cells and Their Niches.
A Wilson, Ann. N.Y. Acad. Sci. 1106: 64–75, 2007 PMID: 17442778
3. Hematopoietic Stem Cells Reversibly Switch from Dormancy to Self-Renewal during Homeostasis and Repair. A Wilson, Cell 135:1118–1129, 2008 PMID: 19062086
4. Intermediate-term hematopoietic stem cells with extended but time-limited reconstitution potential.
P Benveniste, Cell Stem Cell. 6:48–58, 2010 PMID: 20074534
5. Combined Single-Cell Functional and Gene Expression Analysis Resolves Heterogeneity within Stem Cell Populations. NK Wilson, Cell Stem Cell 16:712–724, 2015 PMID: 26004780
6. N-Cadherin-Expressing Bone and Marrow Stromal Progenitor Cells Maintain Reserve Hematopoietic Stem Cells. M Zhao, Cell Reports 26:652–669, 2019 PMID: 30650358
7. BloodSpot 3.0: a database of gene and protein expression data in normal and malignant haematopoiesis
MH Gíslason, Nucleic Acids Research, 52(D1):D1138-D1142, 2024 PMID: 37933860
8. CD49b identifies functionally and epigenetically distinct subsets of lineage biased hematopoietic stem cells. E Somuncular, Stem Cell Reports 17:1546-1560, 2022 PMID: 35714596

References

1. Dykstra, B., Olthof, S., Schreuder, J., Ritsema, M. & Haan, G. De. Clonal analysis reveals multiple functional defects of aged murine hematopoietic stem cells. *Journal of Experimental Medicine* **208**, (2011).
2. Yamamoto, R. *et al.* Large-Scale Clonal Analysis Resolves Aging of the Mouse Hematopoietic Stem Cell Compartment. *Cell Stem Cell* **22**, (2018).
3. Jang, G. *et al.* Stem cell decoupling underlies impaired lymphoid development during aging. *Proc Natl Acad Sci U S A* **120**, (2023).
4. Su, T.-Y. *et al.* Aging is associated with functional and molecular changes in distinct hematopoietic stem cell subsets. *Nat Commun* **15**, 7966 (2024).
5. Hérault, L. *et al.* Single-cell RNA-seq reveals a concomitant delay in differentiation and cell cycle of aged hematopoietic stem cells. *BMC Biol* **19**, 19 (2021).
6. fluentbio.com. Scalable and Cost-Effective scRNA-seq with the PIPseq™ v4.0 Chemistry.
7. Macosko, E. Z. *et al.* Highly Parallel Genome-wide Expression Profiling of Individual Cells Using Nanoliter Droplets. *Cell* **161**, 1202–1214 (2015).
8. Zheng, G. X. Y. *et al.* Massively parallel digital transcriptional profiling of single cells. *Nat Commun* **8**, 14049 (2017).
9. Clark, I. C. *et al.* Microfluidics-free single-cell genomics with templated emulsification. *Nat Biotechnol* **41**, 1557–1566 (2023).
10. Andreatta, M., Berenstein, A. J. & Carmona, S. J. scGate: marker-based purification of cell types from heterogeneous single-cell RNA-seq datasets. *Bioinformatics* **38**, 2642–2644 (2022).
11. Gnani, D. *et al.* An early-senescence state in aged mesenchymal stromal cells contributes to hematopoietic stem and progenitor cell clonogenic impairment through the activation of a pro-inflammatory program. *Aging Cell* **18**, (2019).
12. Poisa-Beiro, L. *et al.* A Senescent Cluster in Aged Human Hematopoietic Stem Cell Compartment as Target for Senotherapy. *Int J Mol Sci* **26**, 787 (2025).
13. Hou, J. *et al.* Aged bone marrow macrophages drive systemic aging and age-related dysfunction via extracellular vesicle-mediated induction of paracrine senescence. *Nat Aging* **4**, 1562–1581 (2024).
14. Qiu, J., Menon, V., Tzavaras, N., Liang, R. & Ghaffari, S. Protocol to identify and analyze mouse and human quiescent hematopoietic stem cells using flow cytometry combined with confocal imaging. *STAR Protoc* **3**, 101828 (2022).
15. Kucinski, I. *et al.* A time- and single-cell-resolved model of murine bone marrow hematopoiesis. *Cell Stem Cell* **31**, 244-259.e10 (2024).
16. Rodriguez-Fraticelli, A. E. *et al.* Clonal analysis of lineage fate in native haematopoiesis. *Nature* **553**, 212–216 (2018).
17. Desterke, C., Bennaceur-Griscelli, A. & Turhan, A. G. EGR1 dysregulation defines an inflammatory and leukemic program in cell trajectory of human-aged hematopoietic stem cells (HSC). *Stem Cell Res Ther* **12**, 419 (2021).
18. Min, I. M. *et al.* The Transcription Factor EGR1 Controls Both the Proliferation and Localization of Hematopoietic Stem Cells. *Cell Stem Cell* **2**, 380–391 (2008).
19. Stoddart, A., Fernald, A. A., Davis, E. M., McNerney, M. E. & Le Beau, M. M. EGR1 Haploinsufficiency Confers a Fitness Advantage to Hematopoietic Stem Cells Following Chemotherapy. *Exp Hematol* **115**, 54–67 (2022).
20. Keenan, A. B. *et al.* ChEA3: transcription factor enrichment analysis by orthogonal omics integration. *Nucleic Acids Res* **47**, W212–W224 (2019).

21. Stoddart, A., Fernald, A. A., Davis, E. M., McNerney, M. E. & Le Beau, M. M. EGR1 Haploinsufficiency Confers a Fitness Advantage to Hematopoietic Stem Cells Following Chemotherapy. *Exp Hematol* **115**, 54–67 (2022).

REVIEWER COMMENTS

Reviewer #1 (Remarks to the Author):

In their revised manuscript describing PURE-seq, the authors largely addressed the previous comments by this Reviewer. The additional data on circulating tumor cells is another solid showcase of the methodology. Still, the CNV analysis appears to be very noisy (healthy cells also appear to have multiple CNVs, which seems highly unlikely). This is likely attributable to low data sparsity. Some type of ground truth data of the primary melanoma would have been useful; otherwise, the low confidence nature of the CNV analysis should be transparently discussed.

We thank the reviewer for their thoughtful comment on the CNV analysis. We have expanded the discussion on CNV analysis in the Discussion by adding: "Of note, while the inferCNV analysis supported the malignant identity of CTCs, RNA-based CNV inference is inherently noisy, and the alterations in reference cells likely reflect data sparsity rather than true genomic changes. Future integration of scRNA-seq or DNA-seq data of the primary melanoma could improve CNV accuracy and confidence."

Reviewer #2 (Remarks to the Author):

the authors have addressed my previous comments adequately. The manuscript describes a very useful new technique that can be applied to the study of small populations of cells in vivo and in vitro.

We thank the reviewer for their positive appraisal of our manuscript.

Reviewer #3 (Remarks to the Author):

This study characterizes and analyzes transcriptomes in an enriched precursor cell fraction from young, middle-aged and old mice. Age-related change in prevalence of Egr-1 transcripts is highlighted. A particular configuration of FACS sorting, cell capture and emulsion encapsulation, branded as 'PURE-seq', is also highlighted.

The submitted revision improves clarity of writing, softens a number of inadequately supported claims, and adds helpful context from the published literature. The manuscript however remains descriptive without functional testing of identities or proposed mechanisms. It is difficult to discern substantive advance over copious published work in this area, in technology, concept or understanding of precursor cell mechanisms.

We thank the reviewer for the feedback and for acknowledging the improvements made to the clarity and contextualization of our manuscript.